# Balanced Federated Clustering via Anchor-Guided Dual Label Learning

## Abstract

Although the $\ell_{2,q}$-norm has been widely used in robust feature extraction and sparse modeling, its potential in promoting clustering balance has long been overlooked. This paper theoretically reveals the inherent ability of the $\ell_{2,q}$-norm to encourage balanced clustering, and proposes a federated multi-view clustering framework that incorporates it as a balance-aware regularizer. While preserving data privacy, the framework employs an efficient optimization strategy to learn a single label matrix, from which both anchor and sample labels can be inferred. The anchor labels then guide sample clustering, leading to improved clustering performance and robustness.

## 1 Introduction

Clustering is a fundamental task in machine learning and data analysis, aiming to group similar data points into the same clusters without relying on labeled data. With the growing emphasis on data privacy, Federated Multi-view Clustering (FedMVC) Yang et al. (2019); Huang et al. (2024) has emerged as an important research paradigm. FedMVC aims to perform efficient unsupervised clustering when data is distributed across multiple clients and cannot be centrally stored or shared, thereby fully leveraging multi-view information without accessing the raw data.

To enhance clustering performance, researchers have proposed various regularization strategies to capture underlying structural information and improve model robustness. For example, graph Laplacian regularization Belkin & Niyogi (2003); Zhao et al. (2022) helps preserve the local manifold structure of data; the nuclear norm Liu et al. (2010a) facilitates the discovery of low-rank structures; and the $\ell_1$-norm is widely used for sparse modeling Tibshirani (1996); Elhamifar & Vidal (2013). Among these tools, the $\ell_{2,q}$-norm has gained popularity in the machine learning community for its ability to promote row sparsity and enhance robustness to noise and outliers.

Nevertheless, existing methods still face significant challenges in the following two aspects:

**Underutilized clustering balance potential.** Although the $\ell_{2,q}$-norm is widely used to promote row sparsity, helping to identify representative features or structures, most existing methods overlook its potential in modeling the distribution structure among samples. In fact, this norm can naturally guide samples to be more evenly distributed across different clusters, effectively preventing degenerate clustering solutions dominated by certain clusters.

**Lack of end-to-end method.** Most existing federated multi-view clustering methods are built upon a two-step framework, where local feature representation learning is first performed on each client, followed by global clustering on the server. However, since the feature extractors are not jointly optimized with the clustering objective, the resulting cluster assignments may be suboptimal.

To this end, we design a balance-aware federated multi-view clustering framework based on dual-label learning, which features the following key components:

- **Balance-aware regularization.** Clustering balance is crucial to avoid degenerate solutions with highly imbalanced cluster sizes. We discover that the $\ell_{2,q}$-norm can achieve balanced clustering and provide a theoretical analysis to support this. Additionally, we propose an efficient optimization method to solve the related problem, ensuring that anchors are evenly distributed across clusters and provide equitable supervision.

- **End-to-end dual-label learning.** We formulate a joint regression model that explicitly links the anchor graph, sample-indicator matrix, and anchor-indicator matrix through latent probabilities. This enables end-to-end learning of both label sets, with sample updates informed by anchor predictions and vice versa, fostering mutual reinforcement.
- **Federated optimization with tensor aggregation.** We embed the above components into a federated learning protocol with adaptive view weights. Clients perform local updates on their own views, and the server aggregates per-view sample-label tensors using a Schatten-$p$-norm penalty, aligning local models without exposing raw data.

## 2 RELATED WORK

### 2.1 $\ell_{2,q}$-NORM IN CLUSTERING

The $\ell_{2,1}$-norm (a special case of the $\ell_{2,q}$-norm) has been widely applied in feature selection, subspace learning, and sparse modeling due to its robustness to noise and outliers as well as its ability to enforce row sparsity. In feature selection, Nie et al. (2010) proposed an efficient and robust framework by jointly minimizing the $\ell_{2,1}$-norm in both the loss and regularization terms. Zhang et al. (2015) further combined the $\ell_{2,1}$-norm with the Fisher discriminant criterion to enhance inter-class separability, resulting in more discriminative and compact feature representations. In multi-task learning, the $\ell_{2,1}$-norm is used to encourage multiple tasks to share the same set of features. Argyriou et al. (2006) pioneered a multi-task feature learning model that imposed $\ell_{2,1}$-regularization on the weight matrix to obtain a consistent sparse structure across tasks. Compared with Euclidean norms, the $\ell_{2,1}$-norm is rotation-invariant and exhibits strong resistance to outliers, making it widely used in sparse learning scenarios. Ding et al. (2006) proposed R1-PCA, which replaces the traditional squared $\ell_2$-reconstruction error in PCA with a rotation-invariant $\ell_1$-norm to improve robustness against outliers. Nie et al. (2021) further proposed a non-greedy $\ell_{2,1}$-norm maximization PCA framework, which aligns better with PCA's original goal of minimizing reconstruction error, while offering stronger robustness, scalability to large datasets, and theoretical convergence guarantees. This method has demonstrated excellent performance on real-world data.

### 2.2 FEDERATED MULTI-VIEW CLUSTERING

With the growing demand for privacy preservation, FedMVC is emerging as a promising research paradigm. Recent federated clustering methods are mostly built upon two-stage frameworks. Specifically, Qiao et al. (2023) constructs an approximate kernel matrix in a privacy-preserving manner, followed by spectral clustering on the server; Huang et al. (2022) performs non-negative matrix factorization locally on each client and then aggregates the results via global K-means; building upon these ideas, Hu et al. (2023) further extends to a federated multi-view fuzzy K-means method that softly assigns samples to clusters. These approaches decouple feature extraction from clustering and rely on post-processing to produce final labels. Meanwhile, existing aggregation strategies Qiao et al. (2023); Cao et al. (2021) (e.g., simple averaging or voting) fail to effectively reconcile the divergent representations that different clients may learn for the same underlying clusters.

## 3 NOTATIONS

For matrix $\mathbf{X} \in \mathbb{R}^{d_v \times n}$, $x_{ij}$ is $(i,j)$-element of $\mathbf{X}$, $\mathbf{x}_j$ and $\mathbf{x}^i$ are the $j$th column and $i$th row of $\mathbf{X}$ respectively. We use bold calligraphy letters for 3rd-order tensors, $\mathcal{H} \in \mathbb{R}^{n_1 \times n_2 \times n_3}$. The $i$th frontal slice of $\mathcal{H}$ is $\mathbf{H}^{(i)}$. $\overline{\mathcal{H}}$ is the discrete Fourier transform of $\mathcal{H}$ along the third dimension, $\overline{\mathcal{H}} = \text{fft}(\mathcal{H}, [\,], 3)$. Thus, $\mathcal{H} = \text{ifft}(\overline{\mathcal{H}}, [\,], 3)$.

**Definition 3.1** (t-product Kilmer & Martin (2011)). Let $\mathcal{X} \in \mathbb{R}^{n_1 \times m \times n_3}$ and $\mathcal{Y} \in \mathbb{R}^{m \times n_2 \times n_3}$. Their t-product $\mathcal{X} * \mathcal{Y} \in \mathbb{R}^{n_1 \times n_2 \times n_3}$ is defined by

$$\mathcal{X} * \mathcal{Y} = \text{ifft}(\text{bdiag}(\overline{\mathbf{X}}\,\overline{\mathbf{Y}}), [\,], 3),$$

where $\overline{\mathbf{X}} = \text{bdiag}(\overline{\mathcal{X}})$ and $\overline{\mathbf{Y}} = \text{bdiag}(\overline{\mathcal{Y}})$ form block-diagonal matrices of their frontal slices.

**Definition 3.2** (t-SVD Kilmer & Martin (2011)). The tensor singular value decomposition of $\mathcal{Z} \in \mathbb{R}^{n_1 \times n_2 \times n_3}$ is

$$\mathcal{Z} = \mathcal{U} * \mathcal{A} * \mathcal{V}^{\top}$$

where $\mathcal{U} \in \mathbb{R}^{n_1 \times n_1 \times n_3}$ and $\mathcal{V} \in \mathbb{R}^{n_2 \times n_2 \times n_3}$ are orthogonal tensors, and $\mathcal{A} \in \mathbb{R}^{n_1 \times n_2 \times n_3}$ is f-diagonal. Here, $*$ denotes the t-product.

**Definition 3.3.** (Tensor Schatten $p$-norm Gao et al. (2021)) Given $\mathcal{H} \in \mathbb{R}^{n_1 \times n_2 \times n_3}$, $h = min(n_1, n_2)$, the tensor Schatten $p$-norm of $\mathcal{H}$ is

$$\|\mathcal{H}\|_{\circledS\!\mathbb{p}} = (\sum_{i=1}^{n_3} |\overline{\mathcal{H}}^{(i)}|_{\circledS\!\mathbb{p}}^p)^{\frac{1}{p}} = (\sum_{i=1}^{n_3} \sum_{j=1}^{h} \sigma_j(\overline{\mathcal{H}}^{(i)})^p)^{\frac{1}{p}}, \tag{1}$$

where $\sigma_j(\overline{\mathcal{H}}^{(i)})$ denotes the $j$th singular value of $\overline{\mathcal{H}}^{(i)}$. By selecting an appropriate $p$, the Schatten $p$-norm provides a more accurate approximation of the rank function.

**Definition 3.4.** For a matrix $\mathbf{Y} \in \mathbb{R}^{n_1 \times n_2}$, the $\ell_{2,q}$-norm Liao et al. (2018) ($0 < q < 2$) is defined as:

$$\|\mathbf{Y}\|_{2,q} = \sum_{i=1}^{n_1} (\sum_{j=1}^{n_2} y_{ij}^2)^{q/2}.$$

# 4 METHODOLOGY

## 4.1 BALANCE REGULARIZATION

To achieve a balanced distribution of labels and avoid trivial solutions, we propose the following novel balance regularization term.

$$\max_{Y\mathbf{1}=\mathbf{1}, Y \geq 0} \|\mathbf{Y}^\top\|_{2,q}, \tag{2}$$

which balances spread and sharpness in the label assignments.

**Theorem 4.1.** *Given label matrix $\mathbf{Y} \in \mathbb{R}^{m \times c}$ with $\mathbf{Y}\mathbf{1} = \mathbf{1}$ and $\mathbf{Y} \geq 0$. Let $0 < q < 2$, then*

$$\max_{\mathbf{Y}\mathbf{1}=\mathbf{1}, \mathbf{Y} \geq 0} \|\mathbf{Y}^\top\|_{2,q} \tag{3}$$

*is equivalent to*

$$\max_{\mathbf{Y}\mathbf{1}=\mathbf{1}, \mathbf{Y} \geq 0} (\|\mathbf{Y}\|_F^2)^{q/2},$$
$$s.t. \quad \|\mathbf{y}_1\|_2^2 = \|\mathbf{y}_2\|_2^2 = \cdots = \|\mathbf{y}_c\|_2^2 \tag{4}$$

*where $\mathbf{y}_j$ denotes the $j$-th column of $\mathbf{Y}$. The detailed proof is provided in Appendix A.1.*

**Theorem 4.2.** *Let $m_j$ ($j = 1, \cdots, c$) be the number of samples in the $j$-th cluster and $0 < q < 2$. Then,*

$$\max_{\mathbf{Y}\mathbf{1}=\mathbf{1}, \mathbf{Y} \geq 0} \|\mathbf{Y}^\top\|_{2,q}, \tag{5}$$

*can ensure a balanced cluster distribution, i.e, $m_1 = m_2 = \cdots = m_c$. The detailed proof is provided in Appendix A.2.*

For $\|\mathbf{Y}^\top\|_{2,q}$ is convex when $0 < q < 2$. To maximize it, according to Theorem 4.3, We relax model with its first-order Taylor expansion as:

$$\max_{\mathbf{Y} \geq 0, \mathbf{Y}\mathbf{1}=\mathbf{1}} \text{tr}(\mathbf{D}^\top \mathbf{Y}), \tag{6}$$

where, $\mathbf{D} = \partial\|\mathbf{Y}^\top\|_{2,q}/\partial\mathbf{Y}$.

**Theorem 4.3.** *Let $f(\mathbf{Y})$ be differentiable and convex in $\mathbf{Y}$. At the current iterate $\mathbf{Y}_k$, its first-order Taylor expansion yields the surrogate*

$$\max_{\mathbf{Y}} \text{tr}(\nabla f(\mathbf{Y}_k)^\top \mathbf{Y}).$$

*This relaxation is valid if and only if $f$ is convex. The detailed proof is provided in Appendix A.3.*

The optimization process of equation 6 can be divided into two steps:

**(1) Fix $\mathbf{Y}$ and solve for $\mathbf{D}$:**

$$\mathbf{D} = \frac{\partial\|\mathbf{Y}^\top\|_{2,q}}{\partial\mathbf{Y}} = q\mathbf{Y}\boldsymbol{\Sigma}, \tag{7}$$

where $\boldsymbol{\Sigma} = \text{diag}(\|\mathbf{y}_i\|_2^{q-2})_{i=1}^c$.

**(2) Fix D and solve for Y**:

$$\mathbf{Y}^{(t+1)} = \arg \max_{\mathbf{Y} \geq 0,\, \mathbf{Y}\mathbf{1}=\mathbf{1}} \operatorname{tr}(\mathbf{D}^\top \mathbf{Y}). \tag{8}$$

Since each row of $\mathbf{Y}$ is independent, the $n$-th row $\mathbf{y}_n$ solves

$$\mathbf{y}_n^* = \arg \max_{\mathbf{y} \geq 0,\, \mathbf{y}\mathbf{1}=1} \mathbf{y}\,(\mathbf{d}_n)^\top, \tag{9}$$

where $\mathbf{d}_n$ is the $n$-th row of $\mathbf{D}$.

The solution to equation 9 is a one-hot vector:

$$y_{nj}^* = \begin{cases} 1, & \text{if } j = \arg \max_{j'} d_{nj'}, \\ 0, & \text{otherwise.} \end{cases} \tag{10}$$

Algorithm 1 details this update.

---

**Algorithm 1** Solving equation 2

1: **Input**: cluster number $c$, $0 < q < 2$.
2: **Output**: label matrix $\mathbf{Y} \in \mathbb{R}^{m \times c}$.
3: **while** not converge **do**
4:     Update $\mathbf{D}$ by equation 7;
5:     Update $\mathbf{Y}$ by equation 10;
6: **end while**
7: **Return** label matrix $\mathbf{Y}$.

---

### 4.2 LOCAL LEARNING IN CLIENTS

Following the approach in Zhao et al. (2025), we relate the anchor graph, anchor labels, and sample labels via

$$\mathbf{Y} = \mathbf{S}^\top \mathbf{H}, \tag{11}$$

where $\mathbf{Y} \in \mathbb{R}^{m \times c}$ is the anchor-label matrix, $\mathbf{S} \in \mathbb{R}^{n \times m}$ is the anchor graph, and $\mathbf{H} \in \mathbb{R}^{n \times c}$ contains the sample labels. By optimizing only $\mathbf{H}$, we recover both sample and anchor labels simultaneously, with the anchors guiding the sample assignments. However, without further constraints on $\mathbf{Y}$, solutions can be non-unique or degenerate. It is important to note that anchor points, selected as representative samples from the original data, play a pivotal role in bridging and guiding the construction of the global clustering structure. If the label distribution of the anchor points becomes trivial or highly skewed, such misleading guidance can severely disrupt the clustering of the samples. Therefore, maintaining a balanced distribution of anchor labels is crucial for improving clustering performance. Therefore, we extend this balance-regularization to federated multi-view clustering by propagating anchor labels on each client. Specifically, on client $v$ we solve

$$\max_{\mathbf{H}^{(v)}\mathbf{1}=\mathbf{1},\, \mathbf{H}^{(v)} \geq 0\, \mathbf{S}^{(v)}\mathbf{1}=\mathbf{1},\, \mathbf{S}^{(v)} \geq 0} \|\mathbf{H}^{(v)^\top} \mathbf{S}^{(v)}\|_{2,q} - \beta \|\mathbf{B}^{(v)} - \mathbf{S}^{(v)}\|_F^2, \tag{12}$$

where $\mathbf{H}^{(v)} \in \mathbb{R}^{n \times c}$ are sample labels, $\mathbf{B}^{(v)} \in \mathbb{R}^{n \times m}$ is the raw anchor graph, and $\mathbf{S}^{(v)}$ its denoised version. The term $\|\cdot\|_{2,q}$ enforces discriminative anchor labels ($0 < q < 2$), while $\beta$ controls denoising strength.

### 4.3 GLOBAL FUSION IN SERVER

After collecting $\{\mathbf{H}^{(v)}\}_{v=1}^V$, the server assembles them into a third-order tensor $\hat{\mathcal{H}}$ (see Figure 1) and enforces low rank along the client mode via a tensor Schatten-$p$ norm. The fusion is

$$\min_{\hat{\mathbf{H}}^{(v)}\mathbf{1}=\mathbf{1},\, \hat{\mathbf{H}}^{(v)} \geq 0,\, \alpha_{(v)} \geq 0,\, \sum_{v=1}^V \alpha_{(v)}=1} \sum_{v=1}^V \frac{1}{\alpha_{(v)}} \|\hat{\mathbf{H}}^{(v)} - \mathbf{H}^{(v)}\|_F^2 + \lambda \|\hat{\mathcal{H}}\|_{\circledS_p}^p, \tag{13}$$

where $\alpha_{(v)}$ weights each client adaptively, balancing heterogeneous data quality.

*Remark* 4.4. The tensor Schatten-$p$ regularizer captures complementary clustering information across clients Gao et al. (2021); Xia et al. (2022). In Figure 1, the $c$th frontal slice $\Delta^{(c)}$ shows sample–cluster affinities across clients. Low-rank enforcement ensures consistency while preserving complementary structures in heterogeneous label assignments.

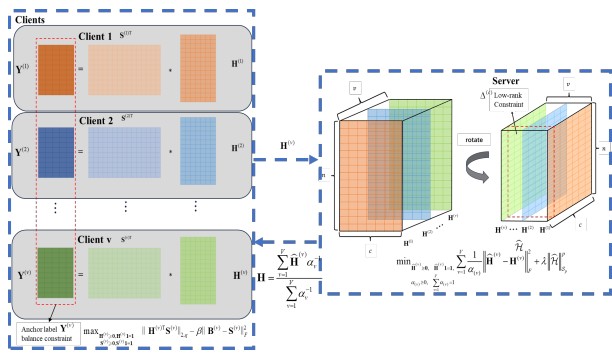

Figure 1: Overview of the proposed framework. Each client performs dual-label learning based on its local anchor graph and uploads sample labels. The server fuses and updates global labels, then sends them back to the clients. This alternating optimization continues until convergence.

## 4.4 CLIENT-SIDE OPTIMIZATION

The client-side optimization is formulated in Eq. equation 12. We solve it via the Alternating Direction Method of Multipliers (ADMM), augmented by a first-order Taylor expansion.

**1. Update of $\mathbf{H}^{(v)}$ with fixed $\mathbf{S}^{(v)}$.** With $\mathbf{S}^{(v)}$ held constant, problem equation 12 reduces to

$$\max_{\mathbf{H}^{(v)} \geq 0, \ \mathbf{H}^{(v)}\mathbf{1}=\mathbf{1}} \|\mathbf{H}^{(v)\top} \mathbf{S}^{(v)}\|_{2,q}. \tag{14}$$

Since each client's data is disjoint, this decouples into $V$ independent subproblems. We approximate equation 14 by linearizing the objective via Theorem 4.3:

$$\mathbf{D}^{(v)} = \frac{\partial \|(\mathbf{Y}^{(v)})^\top\|_{2,q}}{\partial \mathbf{H}^{(v)}} = q \, \mathbf{S}^{(v)}\mathbf{S}^{(v)\top}\mathbf{H}^{(v)} \, \mathbf{\Sigma}^{(v)}, \tag{15}$$

where $\mathbf{\Sigma}^{(v)} = \mathrm{diag}(\|\mathbf{y}_i^{(v)}\|_2^{q-2})_{i=1}^c$.

$$h_{nj}^{(v)*} = \begin{cases} 1, & \text{if } j = \arg \max_{j' \in \{1,\dots,c\}} d_{nj'}^{(v)}, \\ 0, & \text{otherwise.} \end{cases} \tag{16}$$

**(2) Solution for $\mathbf{S}^{(v)}$ with fixed $\mathbf{H}^{(v)}$.** Under this setting, problem equation 12 reduces to:

$$\max_{\mathbf{S}^{(v)} \geq 0, \mathbf{S}^{(v)}\mathbf{1}=\mathbf{1}} \|(\mathbf{H}^{(v)})^\top \mathbf{S}^{(v)}\|_{2,q} \ - \ \beta \|\mathbf{B}^{(v)} - \mathbf{S}^{(v)}\|_F^2. \tag{17}$$

We solve equation 17 iteratively via Theorem 4.3. Define

$$\mathbf{G}^{(v)} = \frac{\partial \|(\mathbf{Y}^{(v)})^\top\|_{2,q}}{\partial \mathbf{Y}^{(v)}} = q \, (\mathbf{S}^{(v)\top} \mathbf{H}^{(v)} \, \mathbf{\Sigma}^{(v)}), \tag{18}$$

accordingly, $\mathbf{S}^{(v)}$ is updated by solving the following equation according to Theorem 4.3.

$$\max_{\mathbf{S}^{(v)} \geq 0, \mathbf{S}^{(v)}\mathbf{1}=\mathbf{1}} \mathrm{tr}(\mathbf{G}^{(v)\top} \mathbf{S}^{(v)\top}\mathbf{H}^{(v)}) \ - \ \beta \|\mathbf{B}^{(v)} - \mathbf{S}^{(v)}\|_F^2, \tag{19}$$

which is equivalently written as

$$\min_{\mathbf{S}^{(v)} \geq 0, \mathbf{S}^{(v)}\mathbf{1}=\mathbf{1}} \mathrm{tr}(\mathbf{S}^{(v)\top}\mathbf{S}^{(v)} - (2\mathbf{B}^{(v)} + \tfrac{1}{\beta} \, \mathbf{H}^{(v)}\mathbf{G}^{(v)\top})^\top \mathbf{S}^{(v)}). \tag{20}$$

Since equation 20 decouples by rows of $\mathbf{S}^{(v)}$, the $n$-th row is obtained by

$$\min_{\mathbf{s}_n^{(v)} \geq 0, \mathbf{s}_n^{(v)}\mathbf{1}=\mathbf{1}} \|\mathbf{s}_n^{(v)} \ - \ (\mathbf{b}_n^{(v)} + \tfrac{1}{2\beta} \, \mathbf{H}^{(v)} \mathbf{g}_n^{(v)\top})\|_2^2. \tag{21}$$

We use the projection onto the simplex Duchi et al. (2008) to solve the above problem, with the detailed algorithm provided in Appendix A.4. Algorithm 2 shows the client-side optimization process.

---

**Algorithm 2** Client-side Optimization

---

1: **Input**: Data matrices $\{\mathbf{X}^{(v)}\}_{v=1}^{V} \in \mathbb{R}^{n \times d_v}$ on $V$ clients, each client construct anchor graphs $\{\mathbf{B}^{(v)}\}_{v=1}^{V} \in \mathbb{R}^{n \times m_v}$ by Xia et al. (2022), sample label $\mathbf{H} \in \mathbb{R}^{n \times c}$ form server.
2: **Output**: Client-side labels $\mathbf{H}^{(v)}$.
3: ▷ on $v$-th client $C_v$
4: **for** $v = 1$ to $V$ **do**
5:    Initialize $\mathbf{H}^{(v)} = \mathbf{H}$
6:    **while** not converged **do**
7:        Update $\mathbf{S}^{(v)}$ by Algorithm 1
8:        Update $\mathbf{H}^{(v)}$ by solving Algorithm 1
9:    **end while**
10: **end for**
11: Send $\mathbf{H}^{(v)}$ to Server

---

### 4.5 SERVER-SIDE OPTIMIZATION

On the server side, we solve the optimization problem in equation 13 via the Augmented Lagrangian Multiplier (ALM) method. Introducing an auxiliary variable $\mathcal{J}$ and enforcing $\hat{\mathcal{H}} = \mathcal{J}$ yields

$$\min_{\hat{\mathbf{H}}^{(v)} \geq 0, \, \hat{\mathbf{H}}^{(v)}\mathbf{1}=\mathbf{1}, \mathcal{J}} \sum_{v=1}^{V} \frac{1}{\alpha_{(v)}} \|\hat{\mathbf{H}}^{(v)} - \mathbf{H}^{(v)}\|_F^2 + \lambda \|\mathcal{J}\|_{\circledS\!\mathcal{P}}^p + \frac{\mu}{2}\|\hat{\mathcal{H}} - \mathcal{J} + \frac{\mathcal{Q}}{\mu}\|_F^2, \tag{22}$$

where $\mathcal{Q}$ is the Lagrange multiplier and $\mu > 0$ is the penalty parameter. We decompose this into three alternating updates.

**(1) Update $\hat{\mathcal{H}}$ with fixed $\mathcal{J}$ and $\{\alpha_{(v)}\}$.**

$$\min_{\hat{\mathbf{H}}^{(v)} \geq 0, \hat{\mathbf{H}}^{(v)}\mathbf{1}=\mathbf{1}} \sum_{v=1}^{V} \frac{1}{\alpha_{(v)}} \|\hat{\mathbf{H}}^{(v)} - \mathbf{H}^{(v)}\|_F^2 + \frac{\mu}{2}\|\hat{\mathcal{H}} - \mathcal{J} + \frac{\mathcal{Q}}{\mu}\|_F^2. \tag{23}$$

Since each view $v$ is independent, this splits into $V$ subproblems:

$$\min_{\hat{\mathbf{H}}^{(v)} \geq 0, \hat{\mathbf{H}}^{(v)}\mathbf{1}=\mathbf{1}} \|\hat{\mathbf{H}}^{(v)} - \mathbf{H}^{(v)}\|_F^2 + \frac{\mu\, \alpha_{(v)}}{2} \|\hat{\mathbf{H}}^{(v)} - \mathbf{M}^{(v)}\|_F^2, \tag{24}$$

where $\mathbf{M}^{(v)} = \mathbf{J}^{(v)} - \frac{\mathbf{Q}^{(v)}}{\mu}$. Each of these can be solved efficiently by projecting onto the probability simplex A.4:

$$\min_{\hat{\mathbf{H}}^{(v)} \geq 0, \hat{\mathbf{H}}^{(v)}\mathbf{1}=\mathbf{1}} \|\hat{\mathbf{H}}^{(v)} - \mathbf{A}^{(v)}\|_F^2, \quad \mathbf{A}^{(v)} = \frac{\mathbf{H}^{(v)} + \frac{\mu\, \alpha_{(v)}}{2}\, \mathbf{W}^{(v)}}{1 + \frac{\mu\, \alpha_{(v)}}{2}}. \tag{25}$$

**(2) Update $\mathcal{J}$ with fixed $\hat{\mathcal{H}}$ and $\{\alpha_{(v)}\}$.**

$$\min_{\mathcal{J}} \frac{\lambda}{\mu}\|\mathcal{J}\|_{\circledS\!\mathcal{P}}^p + \frac{1}{2}\|\hat{\mathcal{H}} - \mathcal{J} + \frac{\mathcal{Q}}{\mu}\|_F^2, \tag{26}$$

which admits the closed-form solution given in Theorem 4.5 Gao et al. (2021):

**Theorem 4.5.** *Let $\mathcal{Z} \in \mathbb{R}^{n_1 \times n_2 \times n_3}$ have the t-SVD $\mathcal{Z} = \mathcal{U} * \mathcal{A} * \mathcal{V}^{\mathsf{T}}$. For*

$$\min_{\mathcal{X}} \frac{1}{2}\|\mathcal{X} - \mathcal{Z}\|_F^2 + \tau \|\mathcal{X}\|_{\circledS\!\mathcal{P}}^p,$$

*the optimal solution is*

$$\mathcal{X}^* = \Gamma_\tau(\mathcal{Z}) = \mathcal{U} * \mathrm{ifft}(\mathbf{P}_\tau(\overline{\mathcal{Z}})) * \mathcal{V}^{\mathsf{T}},$$

*where $\mathbf{P}_\tau(\overline{\mathcal{Z}})$ is the f-diagonal tensor obtained by the Generalized Shrinkage Thresholding algorithm Gao et al. (2021).*

Hence,

$$\mathcal{J}^* = \Gamma_{\frac{\lambda}{\mu}}(\hat{\mathcal{H}} + \frac{\mathcal{Q}}{\mu}). \tag{27}$$

**(3) Update $\{\alpha_{(v)}\}$ with fixed $\hat{\mathcal{H}}$ and $\mathcal{J}$.**

$$\min_{\alpha_{(v)} \geq 0, \sum_{v=1}^{V} \alpha_{(v)} = 1} \sum_{v=1}^{V} \frac{1}{\alpha_{(v)}} \|\hat{\mathbf{H}}^{(v)} - \mathbf{H}^{(v)}\|_F^2. \tag{28}$$

For notational convenience, set $b_v = \|\hat{\mathbf{H}}^{(v)} - \mathbf{H}^{(v)}\|_F^2$. The Lagrangian for equation 28 is

$$\mathcal{L}(\{\alpha_{(v)}\}, \gamma) = \sum_{v=1}^{V} \frac{b_v}{\alpha_{(v)}} - \gamma(\sum_{v=1}^{V} \alpha_{(v)} - 1). \tag{29}$$

Taking the partial derivative with respect to $\alpha_{(v)}$ and setting it to zero, the optimal weights are

$$\alpha_{(v)} = \frac{\sqrt{b_v}}{\sum_{v=1}^{V} \sqrt{b_v}}. \tag{30}$$

Algorithm 3 shows the server-side optimization process. Algorithm 4 summarizes the global optimization.

---

**Algorithm 3** Server-side Optimization

---

**Input:** Client-side labels $\mathbf{H}^{(v)}$.
**Output:** Result $\mathbf{H} \in \mathbb{R}^{n \times c}$
1: $\triangleright$ on Server $S$
2: Initialize $\mathcal{Q} = \mathcal{J} = \mathbf{0}$, $\mu$, $\eta$
3: **while** not converged **do**
4:     Update $\hat{\mathbf{H}}^{(v)}$ by solving equation 25
5:     Update $\mathcal{J}$ by equation 27
6:     Update $\alpha_{(v)}$ by equation 30, update $\mathcal{Q} = \mathcal{Q} + \mu(\mathcal{H} - \mathcal{J})$, $\mu = \eta\mu$
7: **end while**
8: Calculate clustering result $\mathbf{H} = \frac{\sum_{v=1}^{V} \hat{\mathbf{H}}^{(v)} \alpha_v^{-1}}{\sum_{v=1}^{V} \alpha_v^{-1}}$
9: Send $\mathbf{H}$ to Clients

---

**Algorithm 4** Global Optimization

---

**Input:** Data matrices $\{\mathbf{X}^{(v)}\}_{v=1}^{V} \in \mathbb{R}^{n \times d_v}$ on $V$ clients.
**Output:** Result $\mathbf{H} \in \mathbb{R}^{n \times c}$
1: **while** not converged **do**
2:     **Client-side Optimization** on each client by Algorithm 2
3:     **Server-side Optimization** by Algorithm 3
4: **end while**
5: **Return** clustering result $\mathbf{H}$

---

### 4.6 TIME AND SPACE COMPLEXITY ANALYSIS

**Time Complexity:** $\mathcal{O}(V\,n\,m\,d + V\,m^2\,c)$.

**Space Complexity:** $\mathcal{O}(V\,n\,m + 3\,V\,n\,c)$.

The detailed derivation is provided in Appendix A.5.

## 5 EXPERIMENTS

### 5.0.1 DATASETS AND COMPARED METHODS

The descriptions of the datasets and comparison algorithms are provided in Appendix A.6.

## 5.1 Experimental Results

Table 1: Clustering performance on four datasets.

| Methods | BBCSport | | | ORL | | | Yale | | | HAR | | |
|---|---|---|---|---|---|---|---|---|---|---|---|---|
| | ACC | NMI | Purity | ACC | NMI | Purity | ACC | NMI | Purity | ACC | NMI | Purity |
| DiMSC | 0.858 | 0.706 | 0.858 | 0.777 | 0.900 | 0.805 | 0.448 | 0.528 | 0.448 | 0.517 | 0.321 | 0.256 |
| MvLRSSC | 0.628 | 0.404 | 0.646 | 0.635 | 0.800 | 0.668 | 0.440 | 0.480 | 0.450 | 0.493 | 0.535 | 0.534 |
| RMSL | 0.766 | 0.723 | 0.766 | 0.830 | 0.931 | 0.877 | 0.787 | 0.782 | 0.793 | 0.486 | 0.529 | 0.553 |
| GMC | 0.803 | 0.738 | 0.840 | 0.422 | 0.683 | 0.527 | 0.212 | 0.275 | 0.242 | 0.480 | 0.574 | 0.486 |
| MvDGNMF | 0.825 | 0.673 | 0.825 | 0.655 | 0.795 | 0.695 | 0.363 | 0.427 | 0.387 | 0.463 | 0.352 | 0.463 |
| UDBGL | 0.364 | 0.024 | 0.365 | 0.592 | 0.773 | 0.625 | 0.527 | 0.659 | 0.545 | 0.477 | 0.462 | 0.504 |
| FastMICE | 0.439 | 0.111 | 0.454 | 0.787 | 0.904 | 0.822 | 0.624 | 0.570 | 0.654 | 0.567 | 0.495 | 0.567 |
| FedMVL | 0.650 | 0.492 | 0.739 | 0.517 | 0.668 | 0.550 | 0.497 | 0.541 | 0.509 | 0.536 | 0.547 | 0.437 |
| FMVC-IMK | 0.902 | 0.749 | 0.902 | 0.932 | 0.896 | 0.930 | 0.787 | 0.779 | 0.793 | 0.693 | 0.594 | 0.693 |
| TensorFMVC | 0.869 | 0.687 | 0.869 | 0.997 | 0.997 | 0.997 | 0.793 | 0.784 | 0.793 | 0.706 | 0.618 | 0.706 |
| ours | **1** | **1** | **1** | **1** | **1** | **1** | **0.933** | **0.965** | **0.933** | **0.742** | **0.657** | **0.742** |

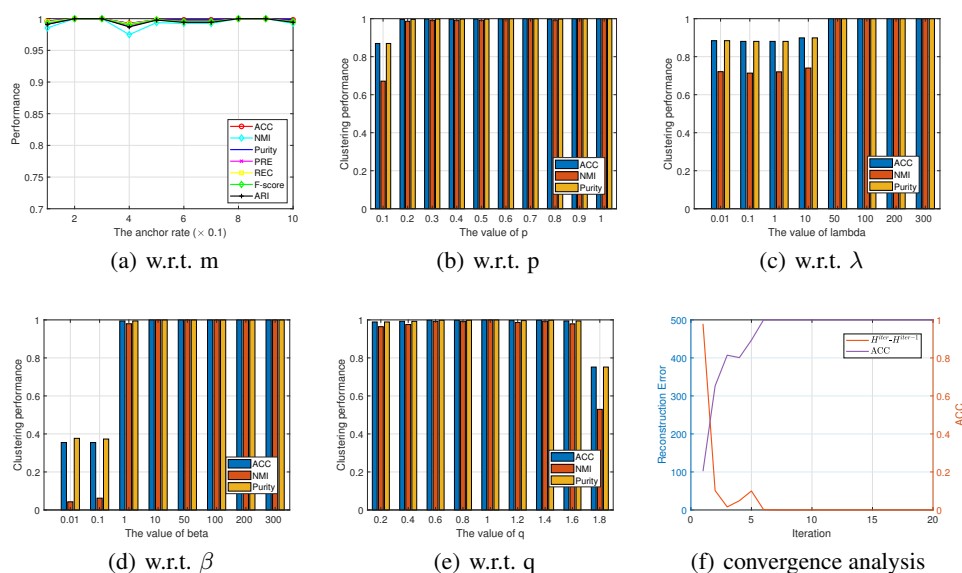

(a) w.r.t. m     (b) w.r.t. p     (c) w.r.t. $\lambda$

(d) w.r.t. $\beta$     (e) w.r.t. q     (f) convergence analysis

Figure 2: Parameter and convergence analysis on BBCsport.

Our method consistently outperforms all competing algorithms across every dataset. Our dual-label mechanism better exploits the structural information of the anchor graph, while the introduced balance regularization ensures model robustness and clustering performance. The adopted Schatten-$p$ norm further enhances the integration of complementary information from different clients.

## 5.2 Parameter Analysis

**(1) Effect of $m$:** We investigated the effect of the number of anchor points ($m$) on clustering performance, as shown in Figure 2(a). The results indicate that increasing the anchor rate does not always improve clustering performance, which may be due to the presence of noise and redundant information in the dataset.

**(2) Effect of $p$:** The parameter $p$ determines the weighting of the tensor's singular values. A smaller $p$ helps better preserve the low-rank structure, while a larger $p$ is more suitable for capturing high-order features. To evaluate the impact of this parameter, the value of $p$ was varied from 0.1 to 1, as shown in Figure 2(b). The results indicate that when $p$ is very small, the clustering performance de-

teriorates. This is likely because excessively enforcing a low-rank structure can suppress important details, thereby degrading the clustering results.

**(3) Effect of $\lambda$:** The impact of the regularization parameter ($\lambda$) on clustering performance was evaluated by varying its value over a range from 0.01 to 300, as shown in Figure 2(c). The results suggested that both overly small and overly large values negatively affected the clustering quality. This indicates that an optimal $\lambda$ is necessary for the effective application of the tensor Schatten $p$-norm, ensuring the best clustering performance.

**(4) Effect of $\beta$:** The impact of the regularization parameter ($\beta$) on clustering performance was evaluated by varying its value over a range from 0.01 to 300, as shown in Figure 2(d). The experimental results demonstrate that when the value is too small, clustering quality deteriorates significantly. This suggests that in such cases, the model fails to effectively suppress noise in the anchor graph, thereby undermining clustering performance. This phenomenon further validates the importance and effectiveness of the anchor graph denoising mechanism in enhancing clustering outcomes.

**(5) Effects of $q$:** The $\ell_{2,q}$-norm parameter ($q$) regulates the relative contributions of individual row $\ell_2$-norms within the overall regularization objective. To assess the impact of this parameter, $q$ was varied within the range $(0, 2)$, as illustrated in Figure 2(f). The experimental results demonstrate that clustering performance deteriorates as $q$ approaches 2. This is because, according to Definition 3.4, as $q$ increases, the objective function gradually approaches the maximization of the squared Frobenius norm. In particular, when $q = 2$, the objective becomes equivalent to maximizing the Frobenius norm, which tends to yield trivial solutions and thus significantly degrades clustering performance. This phenomenon further validates the importance of maintaining balance in the clustering process.

### 5.3 Convergence Analysis

We empirically evaluated the algorithm's convergence by monitoring the difference between the final label matrices at consecutive iterations ($\|\mathbf{H}^{\text{iter}} - \mathbf{H}^{\text{iter}-1}\|_F^2$). Figure 2(f) illustrates the clustering performance during the iterations, where the reconstruction error decreases rapidly and then stabilizes, indicating fast and reliable convergence.

### 5.4 Supplementary Experiments and Analyses

We provide additional experimental results in Appendix A.8, a study on the effect of the tensor rank regularization term in Appendix A.9, an investigation of the impact of the balance constraint in Appendix A.10, a runtime analysis in Appendix A.11, an explanation of the dual-anchor mechanism in Appendix A.12, and a communication complexity analysis in Appendix A.13.

## 6 Conclusion

This paper theoretically proves that maximizing the $\ell_{2,q}$-norm can achieve balanced clustering and proposes a federated clustering framework incorporating this regularization term. By integrating probabilistic modeling, balance-oriented regularization strategies, and privacy-preserving tensor aggregation, the proposed method achieves high-precision clustering in distributed multi-view settings. Detailed theoretical analysis and extensive experiments on multiple benchmark datasets validate the effectiveness and robustness of the proposed approach.

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

# A APPENDIX

## A.1 PROOF OF THEOREM 4.1

*Proof.* According to Definition 3.4, we have

$$\|\mathbf{Y}^\top\|_{2,q} = \sum_{j=1}^{c} \|\mathbf{y}_j\|_2^q = \sum_{j=1}^{c} \left(\|\mathbf{y}_j\|_2^2\right)^{q/2}. \tag{31}$$

Let $\beta_1 = \beta_2 = \cdots = \beta_c = \frac{1}{c}$ and $f(s) = s^{q/2}$. Since $0 < q < 2$, thus $f(s)$ is concave. By Jensen's inequality, we have

$$f\left(\sum_{j=1}^{c} \beta_j \|\mathbf{y}_j\|_2^2\right) \geq \sum_{j=1}^{c} \beta_j f(\|\mathbf{y}_j\|_2^2). \tag{32}$$

Equality holds if and only if $\|\mathbf{y}_1\|_2^2 = \cdots = \|\mathbf{y}_c\|_2^2$.

Simplifying the right-hand side of inequality equation 32, we obtain

$$\sum_{j=1}^{c} \beta_j f(\|\mathbf{y}_j\|_2^2) = \frac{1}{c} \sum_{j=1}^{c} \left(\|\mathbf{y}_j\|_2^2\right)^{q/2} = \frac{1}{c}\|\mathbf{Y}^\top\|_{2,q}. \tag{33}$$

Similarly, simplifying the left-hand side of inequality equation 32, we have

$$f\left(\sum_{j=1}^{c} \beta_j \|\mathbf{y}_j\|_2^2\right) = f\left(\frac{1}{c} \sum_{j=1}^{c} \|\mathbf{y}_j\|_2^2\right) = f\left(\frac{1}{c}\|\mathbf{Y}\|_F^2\right). \tag{34}$$

Substituting Eq. equation 33 and Eq. equation 34 into inequality equation 32, we have

$$f\left(\frac{1}{c}\|\mathbf{Y}\|_F^2\right) \geq \frac{1}{c}\|\mathbf{Y}^\top\|_{2,q}. \tag{35}$$

That is,

$$c\left(\frac{1}{c}\|\mathbf{Y}\|_F^2\right)^{q/2} \geq \|\mathbf{Y}^\top\|_{2,q}. \tag{36}$$

Equality holds if and only if $\|\mathbf{y}_1\|_2^2 = \cdots = \|\mathbf{y}_c\|_2^2$.

Inequality equation 36 indicates that, maximizing the right-hand side of equation 36 is equivalent to maximize the left-hand side of equation 36 with the constraint $\|\mathbf{y}_1\|_2^2 = \cdots = \|\mathbf{y}_c\|_2^2$. Then,

$$\max_{\mathbf{Y1}=\mathbf{1}, \mathbf{Y}\geq 0} \|\mathbf{Y}\|_{2,q} \Leftrightarrow \max_{\mathbf{Y1}=\mathbf{1}, \mathbf{Y}\geq 0} \|\mathbf{Y}\|_F^2, \\ \text{s.t. } \|\mathbf{y}_1\|_2^2 = \cdots = \|\mathbf{y}_c\|_2^2 \tag{37}$$

□

## A.2 PROOF OF THEOREM 4.2

*Proof.* According to Theorem 4.1, optimization problem equation 5 is equivalent to

$$\max_{\mathbf{Y1}=\mathbf{1}, \mathbf{Y}\geq 0} \|\mathbf{Y}\|_F^2, \\ \text{s.t. } \|\mathbf{y}_1\|_2^2 = \cdots = \|\mathbf{y}_c\|_2^2 \tag{38}$$

If there is no constraint $\|\mathbf{y}_1\|_2^2 = \cdots = \|\mathbf{y}_c\|_2^2$, consider the simpler problem

$$\max_{\mathbf{Y1}=\mathbf{1}, \mathbf{Y}\geq 0} \|\mathbf{Y}\|_F^2 = \max_{\mathbf{Y1}=\mathbf{1}, \mathbf{Y}\geq 0} \sum_{i=1}^{m} \sum_{j=1}^{c} y_{ij}^2. \tag{39}$$

Since each row of $\mathbf{Y}$ is independent in equation 39, we can maximize each row separately:

$$\max_{\mathbf{y}_i} \sum_{j=1}^{c} y_{ij}^2 \quad \text{s.t.} \sum_{j=1}^{c} y_{ij} = 1, \quad y_{ij} \geq 0. \tag{40}$$

The objective function $\sum_j y_{ij}^2$ reaches its maximum when the mass 1 is concentrated on a single coordinate. Therefore, any optimal solution of this problem is a one-hot vector, and $\mathbf{Y}$ becomes a one-hot matrix with

$$\|\mathbf{y}_j\|_2^2 = m_j.$$

Here, the 1 in each row can be in any column, and the number of ones in each column can be arbitrary. This results in an unstable and non-unique solution of the model.

Now reintroduce the equality constraint $\|\mathbf{y}_1\|_2^2 = \cdots = \|\mathbf{y}_c\|_2^2$. For one-hot matrices $\mathbf{Y}$, this constraint forces each column to have the same number of ones, thereby ensuring balanced label learning.

Furthermore, by applying Jensen's inequality equation 32, we have

$$f\left(\sum_{j=1}^{c} \frac{1}{c} \|\mathbf{y}_j\|_2^2\right) = f\left(\frac{1}{c} \sum_{j=1}^{c} m_j\right) = f\left(\frac{m}{c}\right)$$

Therefore, the problem equation 5 attains its maximum only when $\mathbf{Y}$ is a discrete matrix and $m_1 = m_2 = \cdots = m_c$. $\qquad\square$

### A.3 PROOF OF THEOREM 4.3

*Proof.* The first-order Taylor expansion of $f(\mathbf{Y})$ at $\mathbf{Y}_k$ is given by:

$$\begin{aligned} F(\mathbf{Y}, \mathbf{Y}_k) &= f(\mathbf{Y}_k) + \langle \nabla f(\mathbf{Y}_k), \mathbf{Y} - \mathbf{Y}_k \rangle \\ &= f(\mathbf{Y}_k) + \text{tr}(\nabla f(\mathbf{Y}_k)^\top (\mathbf{Y} - \mathbf{Y}_k)) \end{aligned} \tag{41}$$

**Necessity:** When approximating $f(\mathbf{Z})$ using its first-order Taylor expansion, the expansion must form a global underestimator of $f(\mathbf{Y})$, i.e.,

$$f(\mathbf{Y}) \geq f(\mathbf{Y}_k) + \text{tr}(\nabla f(\mathbf{Y}_k)^\top (\mathbf{Y} - \mathbf{Y}_k)) \tag{42}$$

Inequality equation 42 is precisely the first-order condition for convexity. Therefore, $f(\mathbf{Y})$ must be a convex function.

**Sufficiency:** If $f(\mathbf{Y})$ is convex, then for any $\mathbf{Y}$, it satisfies the inequality:

$$f(\mathbf{Y}) \geq \text{tr}(\nabla f(\mathbf{Y}_k)^\top \mathbf{Y}) + C \tag{43}$$

where $C = f(\mathbf{Y}_k) - \text{tr}(\nabla f(\mathbf{Y}_k)^\top \mathbf{Y}_k)$ is a constant. Hence, we can replace $f(\mathbf{Y})$ by its lower bound $\text{tr}(\nabla f(\mathbf{Y}_k)^\top \mathbf{Y}) + C$. Ignoring the constant term, the optimization problem reduces to:

$$\max \text{tr}(\nabla f(\mathbf{Y}_k)^\top \mathbf{Y}) \tag{44}$$

Therefore, the necessary and sufficient condition for approximating $f(\mathbf{Y})$ via its first-order Taylor expansion is that $f(\mathbf{Y})$ is convex. $\qquad\square$

### A.4 DETAILS OF THE SIMPLEX PROJECTION ALGORITHM

For the following optimization problem:

$$\min_{\mathbf{s} \in \Delta} \|\mathbf{s} - \mathbf{z}\|_2^2, \qquad \Delta = \{\mathbf{s} \geq 0, \ \mathbf{s}^\top \mathbf{1} = 1\}.$$

$\mathbf{s}_n^{(v)}$ is the Euclidean projection of $\mathbf{z}$ onto the probability simplex $\Delta$. A commonly used algorithm Duchi et al. (2008) proceeds as follows:

1. Sort the entries of $\mathbf{z}$ in descending order: $\tilde{z}_{(1)} \geq \tilde{z}_{(2)} \geq \cdots \geq \tilde{z}_{(m)}$.

2. Find the largest index $k : \tilde{z}_{(k)} - \frac{1}{k}(\sum_{i=1}^{k} \tilde{z}_{(i)} - 1) > 0$.

3. Compute the threshold $\tau = \frac{1}{k}(\sum_{i=1}^{k} \tilde{z}_{(i)} - 1)$.

4. Set $s_i = \max(z_i - \tau, 0), \quad i = 1, \ldots, m$.

The resulting $\mathbf{s}$ satisfies $\mathbf{s} \geq 0$ and $\mathbf{s}^\top \mathbf{1} = 1$, and is the solution of equation 21.

### A.5 TIME AND SPACE COMPLEXITY ANALYSIS

**Time Complexity:** Let $V$, $n$, $m$, $d_v$, and $c$ denote the numbers of views, samples, anchors, features per view, and classes, respectively, and define $d = \sum_{v=1}^{V} d_v$. Our algorithm proceeds in two main stages:

1. Construction of the anchor graphs $\{\mathbf{S}^{(v)}\}_{v=1}^{V}$, which requires

$$\mathcal{O}(V\,n\,m\,d + V\,n\,m\,\log m).$$

2. Iterative updates of the consensus matrix $\mathbf{J}$ and the view-specific cluster assignments $\{\mathbf{H}^{(v)}\}_{v=1}^{V}$, which incur

$$\mathcal{O}(V\,n\,c\,\log(Vn) + V^2\,n\,c) \quad \text{and} \quad \mathcal{O}(V\,n\,c + V\,m^2\,c)$$

costs, respectively.

Since $m$, $c$, and $V$ remain relatively small in practice, the overall time complexity is dominated by

$$\mathcal{O}(V\,n\,m\,d + V\,m^2\,c).$$

**Space Complexity:** Storing the anchor graphs $\{\mathbf{S}^{(v)}\}_{v=1}^{V}$ requires $\mathcal{O}(V\,n\,m)$ memory. Each of the cluster indicator tensor $\mathcal{H}$, the consensus tensor $\mathcal{J}$, and the dual variable tensor $\mathcal{Q}$ requires $\mathcal{O}(V\,n\,c)$. Hence, the total space complexity is

$$\mathcal{O}(V\,n\,m + 3\,V\,n\,c).$$

### A.6 DATASETS AND COMPARED METHODS

We choose the following state-of-the-art algorithms to compare with our proposed method: DiMSC Cao et al. (2015), MvLRSSC Brbić & Kopriva (2018), RMSL Li et al. (2019), GMC Wang et al. (2019), MvDGNMF Li et al. (2020), UDBGL Fang et al. (2023), FastMICE Huang et al. (2023), FedMVL Huang et al. (2022), FMVC-IMK Feng et al. (2024a), TensorFMVC Feng et al. (2024b).

Our experiments are executed on six widely-recognized multi-view datasets: BBCSport Greene & Cunningham (2006), ORL Samaria & Harter (1994), Yale Georghiades et al. (2001), HAR Anguita et al. (2013), SentencesNYU v2(RGB-D) Silberman et al. (2012), Vehicle Sensor Duarte & Hu (2004). Table 2 gives a brief description of these datasets.

Table 2: Datasets

| Datasets | samples | views | classes |
|---|---|---|---|
| BBCSport | 544 | 2 | 5 |
| ORL | 400 | 3 | 40 |
| Yale | 165 | 2 | 15 |
| HAR | 10299 | 4 | 6 |
| Vehicle Sensor | 1954 | 4 | 2 |
| RGB-D | 1449 | 2 | 13 |

## A.7 Additional Experimental Details

All experiments are implemented on a standard Windows 10 Server with two Intel (R) Xeon (R) Gold 6230 CPUs 2.1 GHz and 128 GB RAM, MATLAB R2020a.

To quantitatively evaluate clustering performance, we adopt three widely-used metrics: clustering Accuracy (ACC), Normalized Mutual Information (NMI), and Purity. These metrics compare the predicted cluster assignments with the ground-truth labels.

**Accuracy (ACC).** Clustering accuracy measures the proportion of correctly clustered samples after the best one-to-one mapping between predicted clusters and ground-truth classes is found. It is defined as:

$$\text{ACC} = \frac{\sum_{i=1}^{n} \delta(y_i, \mathcal{M}(c_i))}{n}, \tag{45}$$

where $y_i$ and $c_i$ denote the ground-truth label and predicted cluster label of sample $i$, respectively, $\delta(\cdot)$ is the Kronecker delta function, and $\mathcal{M}$ is the optimal mapping function obtained using the Hungarian algorithm.

**Normalized Mutual Information (NMI).** NMI quantifies the mutual dependence between the predicted cluster assignments $C$ and the ground-truth labels $Y$, normalized by the entropy of each. It is defined as:

$$\text{NMI}(Y, C) = \frac{2 \cdot I(Y; C)}{H(Y) + H(C)}, \tag{46}$$

where $I(Y; C)$ denotes the mutual information between $Y$ and $C$, and $H(\cdot)$ represents entropy. NMI ranges from 0 (no mutual information) to 1 (perfect correlation).

**Purity.** Purity assesses the extent to which each cluster contains data points from a single ground-truth class. It is computed as:

$$\text{Purity} = \frac{1}{n} \sum_{k} \max_{j} |C_k \cap Y_j|, \tag{47}$$

where $C_k$ is the set of samples in cluster $k$, and $Y_j$ is the set of samples with ground-truth label $j$. A higher Purity indicates better clustering consistency with the true classes.

**Nentro.** Normalized entropy (Nentro) is defined as follows:

$$\text{Nentro} = -\frac{1}{\log c} \sum_{h=1}^{c} \frac{n_h}{N} \log \frac{n_h}{N}, \tag{48}$$

where $n_h$ is the size of the $h$-th cluster. An Nentro of 1 indicates perfectly balanced clusters and 0 indicates extremely unbalanced clusters.

These metrics together provide a comprehensive evaluation of clustering quality in terms of label consistency and information overlap.

## A.8 Additional Experimental Results

Additional Experimental Results and hyperparameter settings are listed in Table 3 and Table 4.

## A.9 tensor rank constraints study

First, we conducted an ablation study on the tensor rank constraint, and the results are shown in Table 6. The results demonstrate that simply aggregating outputs from individual views without tensor regularization leads to a noticeable decline in clustering accuracy. In contrast, the tensor rank constraint facilitates the integration of multi-view information and enhances the stability and robustness of the model across diverse data distributions.

We also found that the tensor constraint exhibits robustness under non-IID scenarios. We randomly shuffled the samples within each dataset and reconstructed the client-wise data partitions. We then

Table 3: hyperparameter settings on four datasets.

| Methods | BBCSport | | | ORL | | | Yale | | | HAR | | |
|---|---|---|---|---|---|---|---|---|---|---|---|---|
| | $m$ | $p$ | $\lambda$ | $m$ | $p$ | $\lambda$ | $m$ | $p$ | $\lambda$ | $m$ | $p$ | $\lambda$ |
| | $0.2n$ | 1 | 50 | $0.4n$ | 1 | 100 | $n$ | 0.4 | 200 | $0.01n$ | 1 | 50 |
| | $q$ | $\beta$ | $iter$ | $q$ | $\beta$ | $iter$ | $q$ | $\beta$ | $iter$ | $q$ | $\beta$ | $iter$ |
| | 1 | 50 | 8 | 1 | 1 | 11 | 1 | 10 | 9 | 50 | 1 | 21 |
| | ACC | NMI | Purity | ACC | NMI | Purity | ACC | NMI | Purity | ACC | NMI | Purity |
| DiMSC | 0.858 | 0.706 | 0.858 | 0.777 | 0.900 | 0.805 | 0.448 | 0.528 | 0.448 | 0.517 | 0.321 | 0.256 |
| MvLRSSC | 0.628 | 0.404 | 0.646 | 0.635 | 0.800 | 0.668 | 0.440 | 0.480 | 0.450 | 0.493 | 0.535 | 0.534 |
| RMSL | 0.766 | 0.723 | 0.766 | 0.830 | 0.931 | 0.877 | 0.787 | 0.782 | 0.793 | 0.486 | 0.529 | 0.553 |
| GMC | 0.803 | 0.738 | 0.840 | 0.422 | 0.683 | 0.527 | 0.212 | 0.275 | 0.242 | 0.480 | 0.574 | 0.486 |
| MvDGNMF | 0.825 | 0.673 | 0.825 | 0.655 | 0.795 | 0.695 | 0.363 | 0.427 | 0.387 | 0.463 | 0.352 | 0.463 |
| UDBGL | 0.364 | 0.024 | 0.365 | 0.592 | 0.773 | 0.625 | 0.527 | 0.659 | 0.545 | 0.477 | 0.462 | 0.504 |
| FastMICE | 0.439 | 0.111 | 0.454 | 0.787 | 0.904 | 0.822 | 0.624 | 0.570 | 0.654 | 0.567 | 0.495 | 0.567 |
| FedMVL | 0.650 | 0.492 | 0.739 | 0.517 | 0.668 | 0.550 | 0.497 | 0.541 | 0.509 | 0.536 | 0.547 | 0.437 |
| FMVC-IMK | 0.902 | 0.749 | 0.902 | 0.932 | 0.896 | 0.930 | 0.787 | 0.779 | 0.793 | 0.693 | 0.594 | 0.693 |
| TensorFMVC | 0.869 | 0.687 | 0.869 | 0.997 | 0.997 | 0.997 | 0.793 | 0.784 | 0.793 | 0.706 | 0.618 | 0.706 |
| ours | **1** | **1** | **1** | **1** | **1** | **1** | **0.933** | **0.965** | **0.933** | **0.742** | **0.657** | **0.742** |

Table 4: Supplementary experimental results.

| Methods | Vehicle Sensor | | | rgbd | | |
|---|---|---|---|---|---|---|
| | $m$ | $p$ | $\lambda$ | $m$ | $p$ | $\lambda$ |
| | $0.1n$ | 0.5 | 9900 | $0.1n$ | 0.9 | 200 |
| | $q$ | $\beta$ | $iter$ | $q$ | $\beta$ | $iter$ |
| | 0.1 | 600 | 8 | 0.9 | 5 | 20 |
| | ACC | NMI | Purity | ACC | NMI | Purity |
| DiMSC | 0.689 | 0.222 | 0.689 | 0.396 | 0.326 | 0.497 |
| MvLRSSC | 0.567 | 0.061 | 0.567 | 0.390 | 0.324 | 0.505 |
| RMSL | 0.675 | 0.119 | 0.675 | 0.126 | 0.028 | 0.269 |
| GMC | 0.804 | 0.287 | 0.804 | 0.402 | 0.330 | 0.465 |
| MvDGNMF | 0.500 | 0.060 | 0.500 | 0.265 | 0.007 | 0.272 |
| UDBGL | 0.512 | 0.005 | 0.512 | 0.438 | 0.359 | 0.535 |
| FastMICE | 0.514 | 0.009 | 0.516 | 0.418 | 0.326 | 0.495 |
| FedMVL | 0.763 | 0.225 | 0.763 | 0.332 | 0.455 | 0.223 |
| FMVC-IMK | 0.830 | 0.332 | 0.830 | 0.464 | 0.398 | 0.580 |
| ours | **0.998** | **0.982** | **0.998** | **0.621** | **0.689** | **0.768** |

conducted a full set of experiments under this new dataset setting, with the results reported in Table 5. From these results, we observe that the low-rank tensor fusion method still delivers stable and superior performance in the non-IID scenario. This is mainly attributed to the following:

The low-rank constraint automatically aligns the label subspaces across clients. Even when the client data distributions are inconsistent, it can still effectively extract the global shared structure.

The tensorized fusion mechanism further integrates multi-view and multi-client soft-label information. Under non-IID conditions, it avoids amplifying redundant information and produces more consistent global labels.

Therefore, whether the data follow a non-IID or IID distribution, the proposed low-rank tensor fusion strategy consistently enhances global clustering consistency and stability, demonstrating strong generalization ability and robustness.

Table 5: The non-IID result of 3-sources dataset

| Methods | ACC | NMI | Purity |
|---|---|---|---|
| Shuffle 2 View w.o.$\|\mathcal{H}\|^p_{\mathbb{S}p}$ | 0.402 | 0.162 | 0.485 |
| Shuffle 2 View with $\|\mathcal{H}\|^p_{\mathbb{S}p}$ | 0.988 | 0.972 | 0.988 |
| Shuffle 1 View w.o.$\|\mathcal{H}\|^p_{\mathbb{S}p}$ | 0.609 | 0.495 | 0.727 |
| Shuffle 1 View with $\|\mathcal{H}\|^p_{\mathbb{S}p}$ | 0.994 | 0.981 | 0.994 |
| Ours | **1** | **1** | **1** |

## A.10 BALANCE CONSTRAINTS STUDY

To examine the effect of the balance regularization, we compared our proposed balance term with the traditional Frobenius-norm-based one:

$$\max_{\mathbf{H}^{(v)}\mathbf{1}=\mathbf{1},\,\mathbf{H}^{(v)}\geq 0,\,\mathbf{S}^{(v)}\mathbf{1}=\mathbf{1},\,\mathbf{S}^{(v)}\geq 0}\left\|\mathbf{S}^{(v)\top}\mathbf{H}^{(v)}\right\|_F^2 - \beta\left\|\mathbf{B}^{(v)}-\mathbf{S}^{(v)}\right\|_F^2,$$

We also report the normalized entropy (Nentro) Zhong & Ghosh (2003) to measure cluster balance:

$$\text{Nentro} = -\frac{1}{\log c}\sum_{j=1}^{c}\frac{m_j}{m}\log\frac{m_j}{m},$$

where $m_j$ is the size of the $j$-th cluster, $\sum_{j=1}^{c} m_j = m$. Nentro of 1 indicates perfectly balanced clusters, and 0 indicates extremely unbalanced clusters.

As shown in Table 7, our proposed balance regularization consistently outperforms the Frobenius-norm-based method in both clustering accuracy and uniformity of sample distribution. Traditional methods often produce overly large or small clusters, or even empty clusters, which severely undermine performance. By effectively constraining the number of samples per cluster, our balance term ensures more uniform partitions, preserves the representativeness of different clusters, and prevents any single cluster from dominating the model.

We further visualized the clustering results for each dataset, comparing the ground-truth labels, the results obtained by maximizing the $\ell_{2,q}$-norm, and those obtained by maximizing the Frobenius norm, as shown in Figure 3 and Figure 4. These visualizations indicate that, compared with the Frobenius-norm-based approach, our proposed balance regularization consistently produces more uniform cluster distributions. Notably, in imbalanced datasets, maximizing the Frobenius norm often results in empty clusters, highlighting its inability to ensure uniform sample allocation. In contrast, our balance regularization effectively prevents empty clusters and maintains the representativeness of all clusters.

Moreover, by adjusting the hyperparameter $\beta$, which controls the strength of the balance regularization, its influence on clustering can be flexibly tuned: as shown in Figure 3, for datasets with relatively balanced cluster distributions, a smaller $\beta$ can moderately enhance the effect of the balance regularization, further improving cluster uniformity; whereas for highly imbalanced datasets, as shown in Figure 4, a larger $\beta$ can prevent the regularization from overly influencing the results, allowing the final clustering to retain some inherent imbalance consistent with the actual data distribution. This demonstrates that the proposed method is not only suitable for balanced datasets but can also adapt to highly imbalanced scenarios, providing a flexible and robust mechanism to promote equitable clustering.

Overall, both the quantitative Nentro metrics in Table 7 and the visualized cluster distributions in Figure 3 and Figure 4 clearly demonstrate the advantage of our balance regularization in achieving well-balanced clustering outcomes.

## A.11 RUNNING TIME ANALYSIS

We studied the relationship between the anchor ratio $m$ and the algorithm's running time, as shown in Figure 5. The results indicate that using anchor graphs can significantly reduce running time. Moreover, our parameter analysis shows that the anchor ratio $m$ does not have a linear relationship

Table 6: Ablation study results

| Dataset | Method | ACC | NMI | Purity |
|---------|--------|-----|-----|--------|
| 3-sources | w.o. $\mathcal{H}$ | 0.680 | 0.571 | 0.751 |
| | Ours | **1.000** | **1.000** | **1.000** |
| BBCSport | w.o. $\mathcal{H}$ | 0.862 | 0.686 | 0.862 |
| | Ours | **1.000** | **1.000** | **1.000** |
| ORL | w.o. $\mathcal{H}$ | 0.757 | 0.848 | 0.765 |
| | Ours | **1.000** | **1.000** | **1.000** |
| Yale | w.o. $\mathcal{H}$ | 0.533 | 0.556 | 0.539 |
| | Ours | **0.921** | **0.944** | **0.921** |

Table 7: Balance study results

| Category | Dataset | Method | ACC | NMI | Purity | Nentro |
|----------|---------|--------|-----|-----|--------|--------|
| **Imbalanced** **Datasets** | 3-sources | $\|Y\|_F^2$ | 0.692 | 0.428 | 0.692 | 0.651 |
| | | Ours | **1.000** | **1.000** | **1.000** | **0.887** |
| | BBCSport | $\|Y\|_F^2$ | 0.512 | 0.262 | 0.520 | 0.873 |
| | | Ours | **1.000** | **1.000** | **1.000** | **0.940** |
| **Balanced** **Datasets** | ORL | $\|Y\|_F^2$ | 0.632 | 0.810 | 0.632 | 0.934 |
| | | Ours | **1.000** | **1.000** | **1.000** | **1.000** |
| | Yale | $\|Y\|_F^2$ | 0.466 | 0.497 | 0.484 | 0.907 |
| | | Ours | **0.921** | **0.944** | **0.921** | **0.974** |

with clustering performance. This demonstrates that our framework can maintain low computational overhead even as the data size and the number of clients increase, providing both efficiency and robustness in practical applications.

### A.12 DUAL-LABEL MECHANISM

Our method essentially implements balance regularization in the global label learning problem using a concept regression formulation. For regression tasks,

$$\min_{\mathbf{H},\mathbf{Y}} \|\mathbf{S}^\top \mathbf{H} - \mathbf{Y}\|_F^2 - \beta \|\mathbf{Y}^\top\|_{2,q} \quad s.t. \ \mathbf{H} \geq 0, \mathbf{H1} = \mathbf{1}. \tag{49}$$

Where, anchor graph $\mathbf{S} \in \mathbb{R}^{n \times m}$, projection matrix $\mathbf{H} \in \mathbb{R}^{n \times c}$, anchor indicator matrix $\mathbf{Y} \in \mathbb{R}^{m \times c}$.

Like existing regression based clustering methods, model (49) only considers anchor graph as data representation, and neglects probabilistic property of anchor graph itself. It simultaneously needs to optimize two variables.

Moreover, the elements of anchor graph are non-negative, and each row sums to one. Each row can be considered as the probability that the sample belongs to m anchors. We define the stationary Markov random walks of $\mathbf{S}$ as follows Liu et al. (2010b). The one-step transition probability from the $i$-th sample to the $j$-th anchor is

$$p^{(1)}(u_i \mid x_j) = \frac{s_{ji}}{\sum_{i'}^n s_{ji'}}. \tag{50}$$

Similarly, the transition probability from the $j$-th sample point to the $k$-th category is as follows:

$$p^{(1)}(c_k \mid x_j) = \frac{z_{jk}}{\sum_{k'}^c z_{jk'}} = z_{jk}. \tag{51}$$

Where $z_{jk}$ is the weight between the anchor point $u_j$ and the category $c_k$. In addition, the two Markov processes are independent, hence we provide the transition probabilities from anchors to

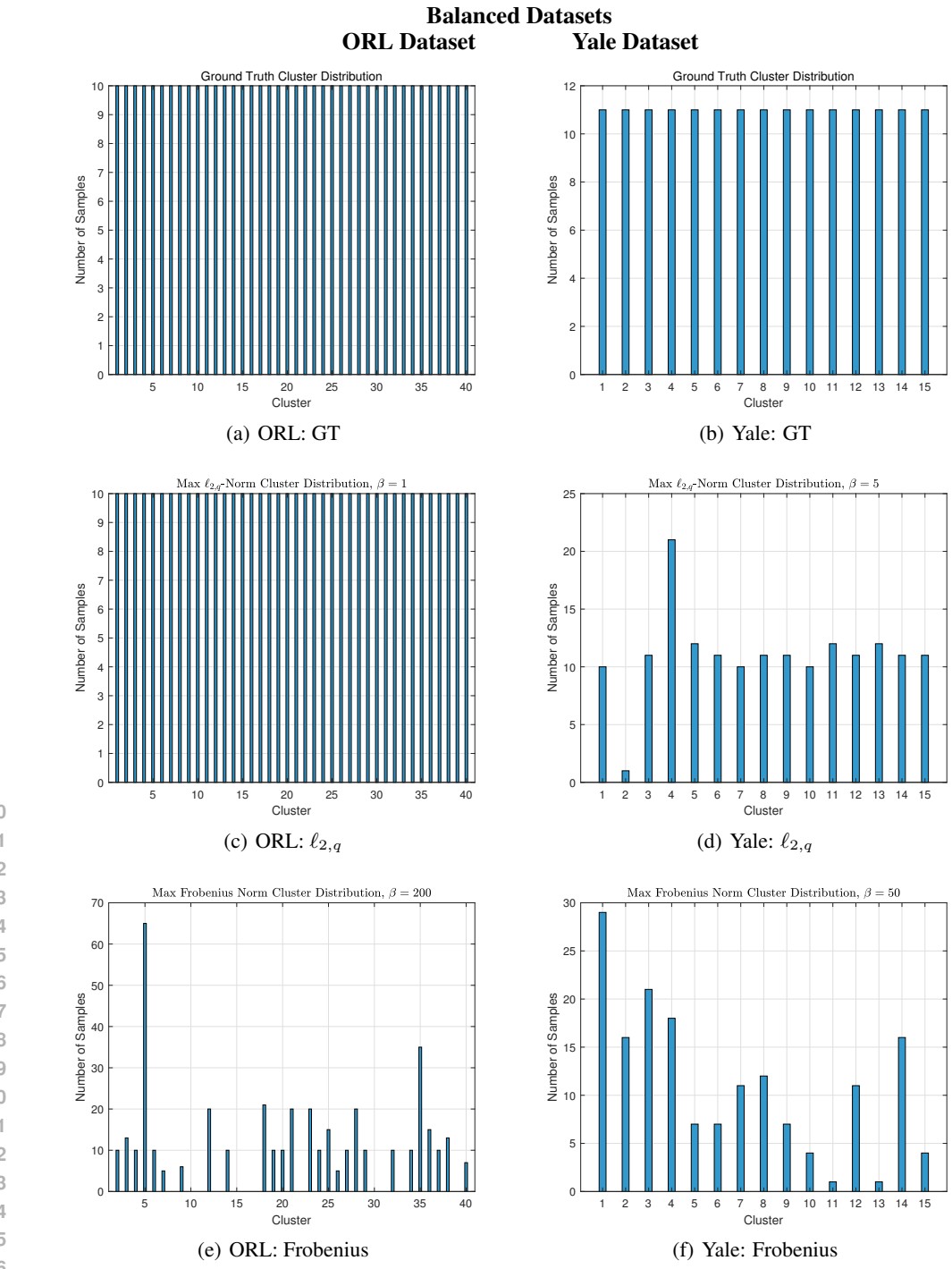

Figure 3: Visualization of clustering results on balanced datasets. Each column corresponds to one dataset, and each row shows results from ground-truth, $\ell_{2,q}$, and Frobenius models, respectively.

categories:

$$p(c_k \mid v_i) = p^{(1)}(u_j \mid v_i)p^{(1)}(c_k \mid u_j) = \frac{s_{ji}z_{jk}}{\sum_{i'}^{n} s_{ji'}}. \tag{52}$$

Rewriting the above equation in matrix form, we have

$$\mathbf{Y} = \mathbf{S}^{\top}\mathbf{H}. \tag{53}$$

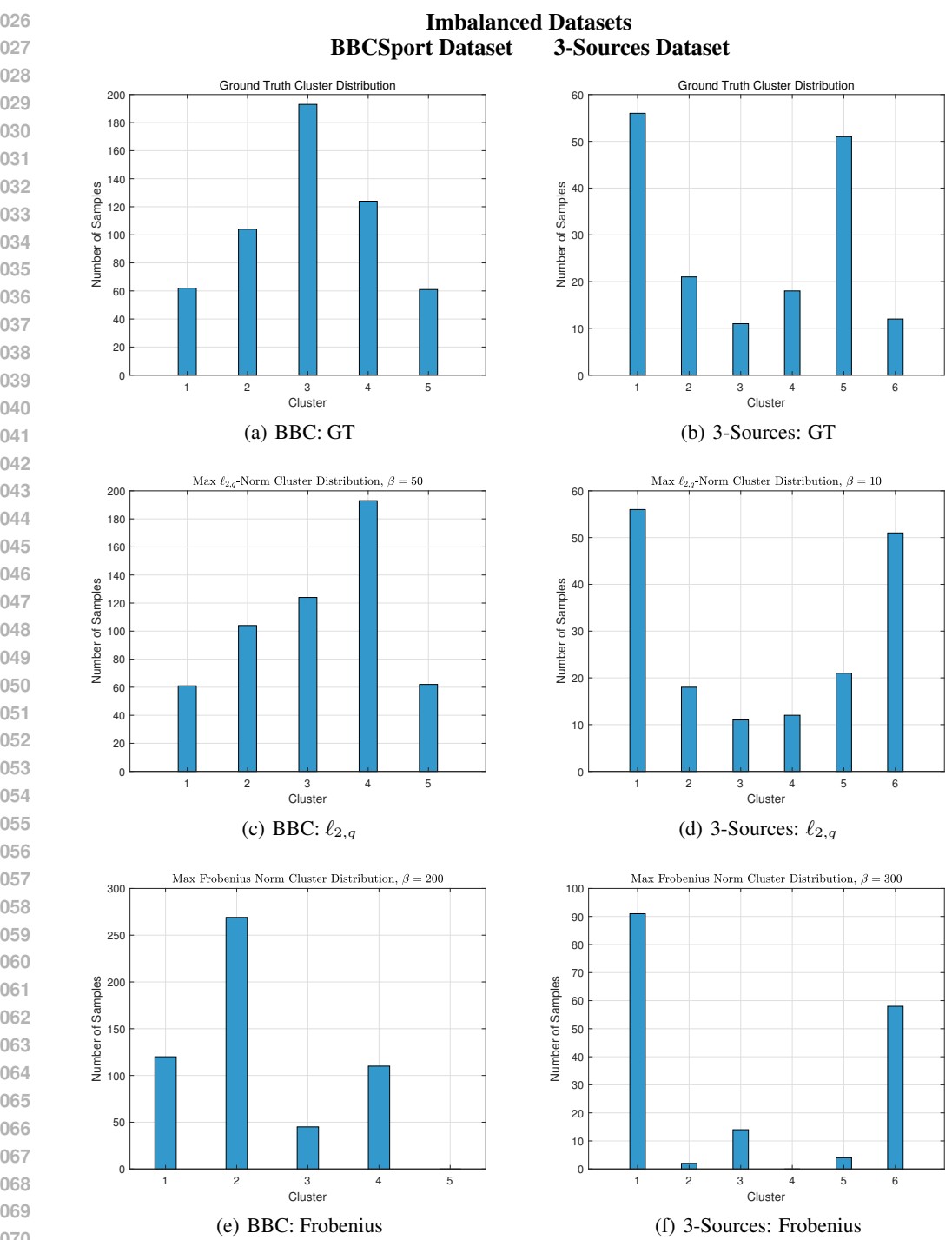

Figure 4: Visualization of clustering results on imbalanced datasets. Each column corresponds to one dataset, and each row shows results from ground-truth, $\ell_{2,q}$, and Frobenius models, respectively.

Figure 6 clearly illustrates the correlation between $n$ sample points, $m$ anchor points, and $c$ categories. Combing model (49) and model (53), we get the concept probabilistic model as

$$\max_{\mathbf{H}} \|\mathbf{H}^\top \mathbf{S}\|_{2,q} \quad s.t. \ \mathbf{H} \geq 0, \mathbf{H1} = \mathbf{1} \tag{54}$$

Anchors serve as representative samples that summarize the global structure. Our proposed bal-

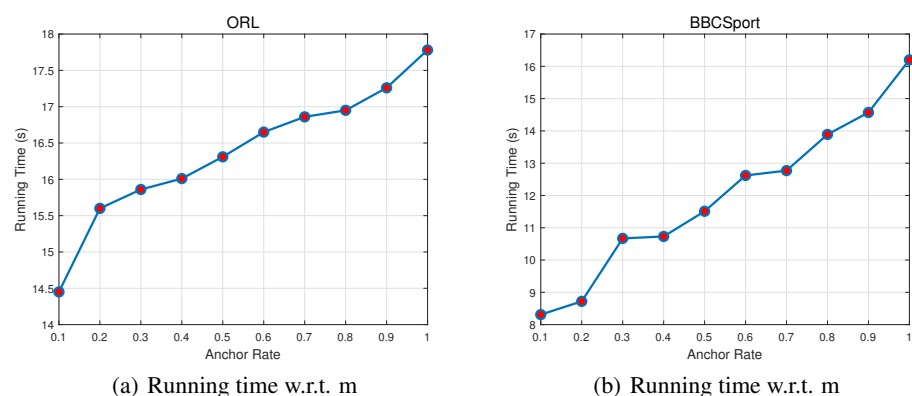

(a) Running time w.r.t. m          (b) Running time w.r.t. m

Figure 5: Running time analysis.

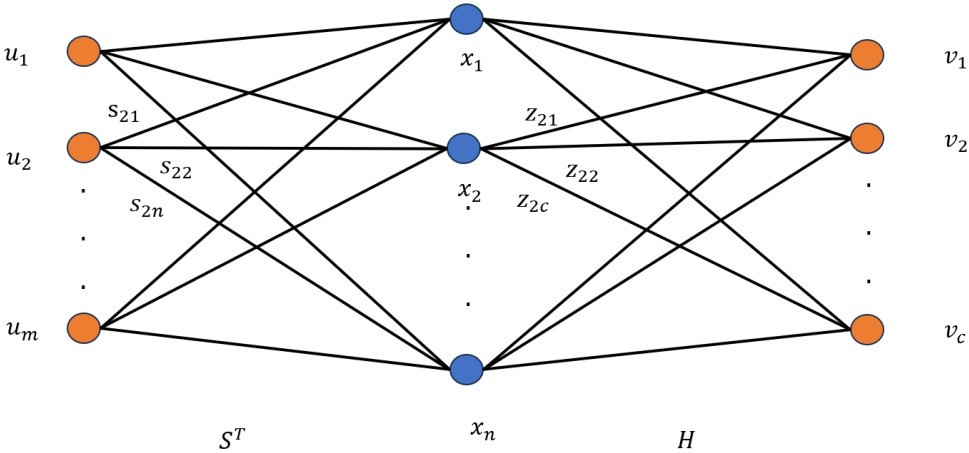

Figure 6: The transition relationship between data points $x_1, x_2, \ldots, x_n$, anchor points $u_1, u_2, \ldots,$ $u_m$ and categories $v_1, v_2, \ldots, v_c$, where $\sum_{j=1}^{c} z_{2j} = \mathbf{1}$.

ancing regularization prevents excessive discrepancies in the anchor distribution, meaning that the selected anchors tend to cover the representative regions of each cluster, which helps ensure that the learned anchor distribution better aligns with the overall data geometry.

Samples continuously refine anchors by providing fine-grained local information. Through the sample-to-anchor update process, the local structures from each client gradually correct and enrich the anchor representations.

The interaction between anchors and samples enhances the stability of multi-client learning. Anchors provide global guidance to samples, while samples iteratively refine the anchors. This bidirectional interaction helps better capture the underlying data distribution.

### A.13 COMMUNICATION COMPLEXITY ANALYSIS

In the proposed federated clustering framework, the communication process is mainly divided into the initialization stage and the iterative optimization stage.

**Initialization stage:** The server broadcasts the initial global label matrix $\mathbf{H} \in \mathbb{R}^{n \times c}$ to all clients, where $n$ is the number of samples and $c$ is the number of clusters. This step is executed only once, with a communication cost of $O(n \cdot c)$.

**Iterative optimization stage:** Each communication round consists of the following steps:

1. **Local client update:** Each client $v$ updates its local label matrix $\mathbf{H}^{(v)} \in \mathbb{R}^{n \times c}$ based on the received global label $\mathbf{H}$ and its local anchor graph $\mathbf{S}^{(v)}$ using Algorithm 2.

2. **Client upload:** All clients send the updated $\mathbf{H}^{(v)}$ to the server. The total upload communication cost is $O(V \cdot n \cdot c)$, where $V$ is the number of clients.

3. **Server aggregation and update:** The server aggregates all $\mathbf{H}^{(v)}$ using Algorithm 3 and updates the global label $\mathbf{H}$.

4. **Server broadcast:** The server broadcasts the updated $\mathbf{H}$ to all clients, with a communication cost of $O(n \cdot c)$.

The total communication cost per iteration is $O(V \cdot n \cdot c)$. Therefore, the overall communication complexity is $O(T \cdot V \cdot n \cdot c)$.

The framework does not use encrypted transmission, but privacy is preserved by exchanging only the label matrices rather than the raw data.

