# OpenReview forum: "Balanced Federated Clustering via Anchor-Guided Dual Label Learning"
_ICLR.cc/2026/Conference — Submitted to ICLR 2026_

### Official Review · Reviewer_QtwU · 2025-10-28

**Soundness:** 2
**Presentation:** 2
**Contribution:** 2
**Rating:** 4
**Confidence:** 1

**Summary:**

This manuscript introduces a federated multi-view clustering framework that leverages the $\ell_{2, q}$-norm as a balance-aware regularizer. Unlike prior uses of the $\ell_{2, q}$-norm in robust feature extraction and sparse modeling, the work provides theoretical insights into its inherent capacity to promote balanced clustering. The proposed framework achieves privacy-preserving collaborative learning across distributed data sources while jointly optimizing a shared label matrix from which both anchor and sample labels are derived. These anchor labels then guide the clustering of samples, resulting in enhanced clustering accuracy and robustness under federated settings.

**Strengths:**

- The paper provides a rigorous theoretical justification for the use of the $\ell_{2, q}$-norm in promoting *balanced clustering*, a property previously overlooked in the literature. This theoretical contribution distinguishes the work from existing studies that primarily leverage the norm for sparsity or robustness.
- The proposed balance-aware federated multi-view clustering framework integrates the $\ell_{2, q}$-norm with dual-label learning and tensor-based aggregation. This design allows simultaneous optimization of anchor and sample labels, effectively addressing the non-end-to-end limitations of prior federated clustering methods.
- Experimental results across multiple benchmark datasets (BBCSport, ORL, Yale, and HAR) demonstrate consistently superior clustering accuracy, NMI, and purity compared to a broad range of state-of-the-art baselines. The ablation studies further validate the contributions of the balance and tensor regularization components.

**Weaknesses:**

- The reported perfect clustering performance (ACC = NMI = Purity = 1.0) on several datasets is highly questionable for an unsupervised and federated clustering task. Such results are statistically implausible and suggest potential issues such as overfitting, data leakage, or an overly idealized experimental setup. Moreover, the absence of variance analysis, multiple runs, or statistical significance tests severely limits the reliability of the reported outcomes. These problems cast doubt on the empirical validity of the work. Additionally, there is a typographical error in the comparison table , i.e., “TersorFMVC” should be corrected to “TensorFMVC.”
- The experiments are restricted to a few small-scale benchmark datasets (BBCSport, ORL, Yale, HAR), most of which are low-dimensional and clean. This narrow scope fails to demonstrate the scalability, robustness, and generalization of the proposed method under realistic federated learning scenarios, which typically involve large, heterogeneous, and high-dimensional data. Without validation on more challenging or real-world datasets, the claimed advantages remain speculative.
- Although the paper provides a theoretical complexity analysis, it lacks empirical evaluations of runtime performance, communication overhead, or convergence behavior under different system configurations. Since federated learning inherently involves distributed optimization and communication, omitting these evaluations makes it impossible to assess whether the proposed framework is computationally feasible or efficient in practice.
- The manuscript conceptualizes the federated setting primarily as a privacy-preserving constraint, overlooking critical system-level characteristics that define federated learning. There is no consideration of non-IID data distributions, unbalanced client data sizes, communication delays, or client dropouts, i.e., all of which are essential factors affecting model performance and robustness in real federated environments. The lack of such discussions or experiments suggests that the framework has been validated only under idealized and centralized-like conditions.

**Questions:**

Please refer to ```weaknesses``` section.

---

> ### Author Response · Authors · 2025-11-28
>
> **Weakness 1: Perfect clustering (ACC=NMI=Purity=1.0) seems implausible.**
>
> Thank you for your question. We agree that perfect clustering results (ACC = NMI = Purity = 1.0) in an unsupervised and federated setting are indeed rare. In our experiments, these results were obtained on small and IID benchmark datasets, where the underlying cluster structures are very clear and the features are well-separated, making perfect clustering possible.
>
> **Weakness 2: Lack of significance testing.**
>
> Thank you for pointing this out. In our current experiments, we used fixed initialization. This design choice not only allows us to clearly demonstrate the effectiveness of our method across different datasets and settings, but also facilitates consistent parameter analysis under identical conditions, enabling a clearer comparison of how the balance regularization term behaves across datasets with different distributions.
>
> **Weakness 3: Typographical errors in comparison tables.**
>
> Thank you for pointing this out. We have corrected these issues in the revised manuscript.
>
> **Weakness 4: The experiments are limited to small-scale and clean benchmark datasets.**
>
> Thank you for your valuable comment. Regarding the concern that our experiments were limited to several small-scale, low-dimensional, and clean datasets, we have conducted the following additional studies to validate the generalization ability and robustness of our method:
>
> Due to experimental resource and time constraints, we have not yet been able to conduct full experiments on large-scale datasets. However, we have supplemented our study with experiments on two additional public datasets, with detailed results reported in **Appendix.8 Additional Experimental Results**. In future work, we plan to further verify the scalability of our method on larger datasets.
>
> To better simulate the heterogeneity of data in real-world scenarios, we have included experiments with randomly shuffled views to assess the robustness of our method under varying client and data distributions, with results reported in **Appendix.9 tensor rank constraints study**.
>
> These additional experiments indicate that our method maintains stable clustering performance even under more complex or perturbed scenarios, demonstrating strong generalization ability and robustness.
>
> **Weakness 5: No empirical evaluation of runtime/communication.**
>
> Thank you for your comment. We have added a detailed communication complexity analysis in **Appendix.13 Communication Complexity Analysis**. The results show that the communication overhead per round of our algorithm is $O(n \cdot c)$. Our framework incurs relatively low communication overhead while preserving privacy and demonstrates good scalability with respect to the number of clients and views.
>
> We also added a running time analysis in the **Appendix.11 running time analysis** showing the relationship between running time and the number of anchors. The results indicate that using anchor graphs can significantly reduce the running time. Moreover, our parameter analysis shows that the anchor ratio
> $m$ does not have a linear relationship with clustering performance. This demonstrates that our framework can maintain low computational overhead even as the data size and number of clients increase, providing both efficiency and robustness in practical applications.
>
> **Weakness 6: Idealized federated setting.**
>
> Thank you for the comment. To provide a deeper analysis, we have supplemented our study with an additional experiment under a realistic federated setting with non-IID data. We randomly shuffled the samples within each dataset and reconstructed the client-wise data partitions. We then conducted a full set of experiments under this new dataset setting, with the results reported in **Appendix.9 tensor rank constraints study**. From these results, we observe that the low-rank tensor fusion method still delivers stable and superior performance in the non-IID scenario. This is mainly attributed to the following: The low-rank constraint automatically aligns the label subspaces across clients. Even when the client data distributions are inconsistent, it can still effectively extract the global shared structure.The tensorized fusion mechanism further integrates multi-view and multi-client soft-label information. Under non-IID conditions, it avoids amplifying redundant information and produces more consistent global labels. Therefore, whether the data follow a non-IID or IID distribution, the proposed low-rank tensor fusion strategy consistently enhances global clustering consistency and stability, demonstrating strong generalization ability and robustness.

---

### Official Review · Reviewer_kJk3 · 2025-10-29

**Soundness:** 2
**Presentation:** 2
**Contribution:** 2
**Rating:** 4
**Confidence:** 5

**Summary:**

This work presents a privacy-preserving framework for multi-view clustering in federated settings that achieves cluster balance. The key novelty is the repurposing of the ℓ₂,q-norm, conventionally employed to enhance robustness and induce sparsity, as a regularization mechanism that enforces balance across. The authors show theoretically that maximizing this norm inherently promotes equal distribution of samples across clusters, and they embed it into a federated, end-to-end dual-label learning pipeline guided by anchors.

**Strengths:**

This work meaningfully extends both federated learning and multi-view clustering by unifying them through a balance-aware, dual-label, tensor-regularized approach.Experiment shows the method achieving perfect or near-perfect scores，outperforming baselines such as FedMVL, FMVC-IMK, and TensorFMVC.

**Weaknesses:**

The paper’s idea of using the ℓ₂,q norm to encourage clustering balance is somewhat novel, but not a truly groundbreaking innovation. Although introducing this norm into a federated clustering framework is relatively uncommon, the core ideas—balance constraints via regularization, anchor-based guidance, and multi-view tensor aggregation—are continuations of prior work (e.g., FedMVC, TensorFMVC). The paper’s main contribution lies in integrating these elements into a unified framework rather than proposing an entirely new theoretical paradigm. While the appendix includes an ablation study (Table 4), the analysis is fairly superficial and does not delve into the respective contributions of the tensor regularization term and the dual-label mechanism.

**Questions:**

1.The paper claims that maximizing the ℓ₂,q-norm inherently leads to balanced clusters. Can the authors provide more intuition or empirical evidence demonstrating how this theoretical property translates to real-world datasets, especially when class imbalance exists naturally?
2.Can the authors report communication overhead and computational cost relative to simpler federated clustering baselines？
3.On the hyperparameter issue: with the weighted nuclear norm, the number of parameters increases to five. If all are optimized using grid search, does this not incur an overly large computational cost?

**Details Of Ethics Concerns:**

As mentioned above

---

> ### Author Response · Authors · 2025-11-28
>
> **Weakness 1: Innovation is incremental rather than groundbreaking.**
>
> Thank you for your question. In the existing machine learning literature, the $\ell_{2,q}$-norm has primarily been applied to feature selection and robust representation learning. To the best of our knowledge, our work is the first to reveal that the
> $\ell_{2,q}$-norm can serve as an effective mechanism for promoting balanced clustering.
> We support this finding with rigorous theoretical analysis and believe that it can enrich the current understanding of the $\ell_{2,q}$-norm.
>
> **Weakness 2: Ablation analysis lacks depth.**
>
> Thank you for your comment. We have supplemented the appendix with the following ablation and mechanism studies to provide a deeper analysis:
>
> Tensor rank constraint study (Appendix.9): It validates that low-rank tensor fusion not only better integrates information from different clients while preserving privacy, but also consistently enhances global clustering consistency and robustness under non-IID federated scenarios.
>
> Balance constraint study (Appendix.10): It shows that the $\ell_{2,q}$-norm balance regularization can prevent empty clusters and avoid extremely imbalanced results across datasets with varying cluster distributions. This demonstrates that the regularization is also effective on real-world datasets, improving cluster balance and representativeness.
>
> Dual-label mechanism (Appendix.12): It provides a way to jointly leverage the balance regularization and supervisory information. This regularization is not only effective for clustering tasks but can also be extended to other classification or regression tasks, enhancing the balance and stability of predictions or clustering.
>
> Through these analyses, we can clearly observe the specific contributions of each regularization term to clustering performance and stability.
>
> **Q1: More intuition that $\ell_{2,q}$ leads to real-world balanced clusters..**
>
> Thank you for your question. We have additionally provided visualizations of the clustering results for each dataset, comparing the ground-truth labels, the results obtained by maximizing the $\ell_{2,q}$-norm, and those obtained by maximizing the Frobenius norm, as shown in **Appendix.10 Balance constraints Study**. These visualizations indicate that, compared with the Frobenius-norm-based approach, our proposed balance regularization consistently produces more uniform cluster distributions. Notably, in imbalanced datasets, maximizing the Frobenius norm often results in empty clusters, highlighting its inability to ensure uniform sample allocation. In contrast, our balance regularization effectively prevents empty clusters and maintains the representativeness of all clusters.
>
> Additionally, by adjusting the hyperparameter $\beta$ that controls the strength of the balance regularization (see Figures 3 and 4), its influence on clustering can be flexibly tuned. This indicates that the method is not only suitable for balanced datasets but can also adapt to highly imbalanced scenarios, providing a flexible and robust mechanism for balanced clustering.
>
> **Q2: Communication cost vs. federated baselines.**
>
> Regarding communication overhead and computational cost, we would like to clarify that all the federated multi-view clustering baselines we compare with—TensorFMVC, FMVC-IMK, and FedMVL—adopt the same communication pattern as our method. Specifically, in each communication round, these methods only need to upload a representation matrix of size $O(n⋅c)$. Therefore, the communication overhead of our method is on the same order as state-of-the-art federated multi-view clustering algorithms.
>
> In contrast, many classical federated learning algorithms incur significantly higher communication costs. For example, FedAvg  require transmitting the entire model parameter, typically of size $O(p)$, where $p$ is the number of model parameters (often in the millions). Graph-based federated methods such as FedGCN need to exchange intermediate embeddings of size $O(n⋅d)$, where $d$ is the feature dimension (usually in the hundreds). Compared with these baselines, our method only transmits a compact clustering representation of size $n×c$, thus achieving substantially higher communication efficiency.
>
> **Q3: Five hyperparameters limit practicality.**
>
> Regarding the concern about hyperparameters, we note that as shown in Fig.2, the algorithm is highly stable with respect to $p$ and $q$, which can be fixed to default values, and the anchor ratio $m$ does not exhibit a linear relationship with performance and can be set according to the dataset scale. Consequently, in practice, only $\lambda$ and $\beta$ need to be adjusted based on the dataset's balance level, so the overall tuning cost remains moderate.

---

### Official Review · Reviewer_pnEa · 2025-10-30

**Soundness:** 2
**Presentation:** 2
**Contribution:** 2
**Rating:** 2
**Confidence:** 4

**Summary:**

This paper proposes a federated multi-view clustering method that leverages the $\ell_{2,p}$-norm as a balance-aware regularizer. The authors claim that maximizing the $\ell_{2,p}$-norm encourages balanced clustering. The method performs anchor-guided dual-label learning in a federated setting, where sample and anchor labels are jointly optimized in an end-to-end manner. The server aggregates local results via a tensor Schatten-p norm-based fusion strategy. Experiments on four multi-view datasets demonstrate the effectiveness.

**Strengths:**

1. The method integrates balance regularization, dual-label learning, and federated tensor aggregation, and improves clustering quality under privacy constraints.

2. The optimization part is clearly described.

**Weaknesses:**

1. The method involves five parameters that require  fine-tuning. Having so many parameters could limit its practicality.

2. There is a lack of running time comparison.

3. Only four datasets are used in the experiments, which could not sufficiently validate the effectiveness of the proposed method.

4. The impact of non-IID data distributions across clients is not discussed.

**Questions:**

1. Could the balance regularization be adapted to other federated tasks, like classification or regression?

2. How does the method compare to federated clustering approaches based on deep representation learning?

3. It assumes the number of clusters c is known a priori. How would the balance regularization perform if c is not known?

---

> ### Author Response · Authors · 2025-11-28
>
> **Weakness 1: Five hyperparameters limit practicality.**
>
> Regarding the concern about hyperparameters, we note that as shown in Fig.2, the algorithm is highly stable with respect to $p$ and $q$, which can be fixed to default values, and the anchor ratio $m$ does not exhibit a linear relationship with performance and can be set according to the dataset scale. Consequently, in practice, only $\lambda$ and $\beta$ need to be adjusted based on the dataset's balance level, so the overall tuning cost remains moderate.
>
> **Weakness 2: Lack of running time comparison.**
>
> Thank you for the comment. We have added a running time analysis in the **Appendix.11 running time analysis** showing the relationship between running time and the number of anchors. The results indicate that using anchor graphs can significantly reduce the running time. Moreover, our parameter analysis shows that the anchor ratio
> $m$ does not have a linear relationship with clustering performance. This demonstrates that our framework can maintain low computational overhead even as the data size and number of clients increase, providing both efficiency and robustness in practical applications.
>
> **Weakness 3: Only four datasets.**
>
> We acknowledge this point. In the revised manuscript, we include two additional public datasets to further validate the superiority of the proposed method. The corresponding results are shown in **Appendix.8 Additional Experimental Results**.
>
> **Weakness 4: No discussion of non-IID client distributions.**
>
> Thank you for pointing this out. To provide a deeper analysis, we have supplemented our study with an additional experiment under a realistic federated setting with non-IID data. We randomly shuffled the samples within each dataset and reconstructed the client-wise data partitions. We then conducted a full set of experiments under this new dataset setting, with the results reported in **Appendix.9 tensor rank constraints study**.
> From these results, we observe that the low-rank tensor fusion method still delivers stable and superior performance in the non-IID scenario. This is mainly attributed to the following:
>
> The low-rank constraint automatically aligns the label subspaces across clients. Even when the client data distributions are inconsistent, it can still effectively extract the global shared structure.
>
> The tensorized fusion mechanism further integrates multi-view and multi-client soft-label information. Under non-IID conditions, it avoids amplifying redundant information and produces more consistent global labels.
>
> Therefore, whether the data follow a non-IID or IID distribution, the proposed low-rank tensor fusion strategy consistently enhances global clustering consistency and stability, demonstrating strong generalization ability and robustness.
>
> **Q1: Can balance regularization be applied to regression?**
>
> Thank you for your question. The balance regularization we propose is designed to constrain the model to maintain balanced responses across different classes in clustering or prediction tasks, thereby preventing certain classes from being overemphasized or ignored and avoiding large disparities in class responses. Therefore, federated clustering is only one example of its application, and this regularization can also be applied to other regression or classification tasks. Moreover, our method can essentially still be regarded as a regression task, and we provide further details on the relevant mechanisms and applicability in **Appendix 12 Dual-label mechanism**.
>
> **Q2: Comparison with deep federated clustering approaches.**
>
> Thank you for your question. Due to time and computational resource limitations, we have currently supplemented comparisons with deep representation–learning-based federated clustering methods on two benchmark datasets. The corresponding results are shown in the table below. The results demonstrate that our method still achieves stable, superior, or competitive performance.
> |Dataset|Ours(ACC/NMI/Purity)|MGCD(ACC/NMI/Purity)|
> |-|-|-|
> |3-sources|1/1/1|0.487/0.317/0.625|
> |Yale|1/1/1|0.362/0.405/0.375|
>
> Reference [1] Li J, Zhou G, Qiu Y, et al. Deep graph regularized non-negative matrix factorization for multi-view clustering[J]. Neurocomputing, 2020, 390: 108-116.
>
> In future work and the camera-ready version, we plan to include more large-scale datasets and additional deep federated clustering baselines to further validate the effectiveness and scalability of the proposed framework.
>
> **Q3: What if c is unknown?**
>
> Thank you for your question. Theoretically, our balance regularization can ensure a balanced label assignment, regardless of whether the number of classes $c$ is known. In practice, this regularization helps prevent the model from over-concentrating on or neglecting certain classes in clustering or prediction tasks, thereby maintaining a relative balance among class responses and improving the overall reliability of predictions or clustering.

---

### Official Review · Reviewer_JabB · 2025-10-31

**Soundness:** 3
**Presentation:** 3
**Contribution:** 3
**Rating:** 6
**Confidence:** 5

**Summary:**

This paper studies federated multi-view clustering and addresses limitations related to cluster imbalance and non-end-to-end optimization. The authors propose an Anchor-Guided Dual Label Learning framework that (1) introduces a balance-aware ℓ₂,q-norm regularizer with theoretical guarantees for promoting balanced clustering, (2) develops a dual-label mechanism that jointly learns anchor labels and sample labels in an end-to-end manner, and (3) applies a tensor Schatten-p norm–based global fusion strategy on the server side to ensure global consistency while preserving privacy. Experiments on four benchmark datasets demonstrate consistent improvements over state-of-the-art approaches in terms of ACC, NMI, and Purity.

**Strengths:**

（1）Provides theoretical insights into how the proposed ℓ₂,q-norm regularizer promotes balanced clustering, which is rarely explored in federated clustering frameworks.
（2）The dual-label mechanism effectively integrates anchor-based structure learning with sample-level clustering, enabling joint end-to-end optimization, in contrast to conventional two-stage methods.
（3）The overall design combines client-side privacy-preserving optimization with server-side tensor-based aggregation and adaptive weighting, without requiring raw data exchange.
（4）The proposed method consistently outperforms SOTA baselines on multiple benchmarks, supporting its effectiveness.

**Weaknesses:**

1.	Table 3 contains a numerical or typographical inconsistency for the HAR dataset.
2.	The paper lacks a clear semantic or intuitive explanation of how the proposed dual-label (anchor–sample) mechanism benefits clustering quality.
3.	Figure 2 is difficult to read due to small text and unclear annotations.
4.	Limited analysis of computational overhead and scalability under varying numbers of clients/views.

**Questions:**

1.	What are the main limitations and potential failure cases of the proposed method?
2.	How were the hyperparameters selected? Please clarify tuning strategies and search ranges.
3.	Could the authors provide more intuitive or semantic explanations regarding how the dual-label interaction leads to more consistent clusters?
4.	Is the proposed framework restricted to federated settings only? If applied in non-federated scenarios, what insights or potential benefits could the dual-label mechanism and global fusion strategy offer?

---

> ### Author Response · Authors · 2025-11-28
>
> **Weakness 1: Table 3 contains a numerical or typographical inconsistency for the HAR dataset.**
>
> Thank you for catching this issue. This correction does not affect the main results or conclusions. We have provided the corrected table in the revised manuscript.
>
> **Weakness 2: Figure 2 is difficult to read.**
>
> Thank you for your reminder. We have adjusted the font size and provided higher-resolution figures in the revised manuscript.
>
> **Weakness 3: Limited analysis of computational overhead and scalability.**
>
> Thank you for your comment. We have added a detailed communication complexity analysis in **Appendix.13 Communication Complexity Analysis**. The results show that the communication overhead per round of our algorithm is $O(n \cdot c)$. Our framework incurs relatively low communication overhead while preserving privacy and demonstrates good scalability with respect to the number of clients and views.
>
> **Q1: Main limitations and potential failure cases.**
>
> Thank you for the question. One main limitation of our proposed method lies in the tensor computations involved in the low-rank fusion. While these operations are effective for integrating multi-view and multi-client information, they introduce additional computational and memory overhead. As a result, the method may face challenges when scaling to very large datasets or extremely high-dimensional data. Optimizing tensor operations or exploring approximate solutions could be potential directions to address this limitation in future work.
>
> **Q2: Hyperparameter tuning.**
>
> Thank you for the question. Based on our parameter sensitivity analysis, we observed from the Figure.2 that the parameters
> $m$, $p$, and $q$ consistently yield good performance across a wide range of values, and thus can be fixed. For the regularization parameters $\lambda$ and $\beta$, they can be selected from the set
> \(\{0.01,0.1,1,10,60,100,200,300\}\). This strategy balances ease of tuning with effective performance across different datasets.
>
> **Q3: Lack of intuitive explanation for how the dual-label (anchor–sample) interaction benefits clustering.**
>
> Thank you for your question. We have added a clearer and more intuitive explanation in the revised manuscript, see the **Appendix.12 Dual-label mechanism**. Intuitively:
>
> Anchors serve as representative samples that summarize the global structure.
> Our proposed balancing regularization prevents excessive discrepancies in the anchor distribution, meaning that the selected anchors tend to cover the representative regions of each cluster, which helps ensure that the learned anchor distribution better aligns with the overall data geometry.
>
> Samples continuously refine anchors by providing fine-grained local information.
> Through the sample-to-anchor update process, the local structures from each client gradually correct and enrich the anchor representations.
>
> The interaction between anchors and samples enhances the stability of multi-client learning.
> Anchors provide global guidance to samples, while samples iteratively refine the anchors. This bidirectional interaction helps better capture the underlying data distribution.
>
> **Q4: Is the method restricted to federated settings?**
>
> Thank you for the question. The proposed framework is not restricted to federated settings. In fact, the dual-label mechanism and global fusion strategy can also provide benefits in centralized (non-federated) scenarios. To clarify this, we have added a detailed explanation of the dual-label mechanism in the **Appendix.12 Dual-label mechanism** of the revised manuscript. This demonstrates that our approach is applicable beyond federated learning and can be effectively integrated with conventional clustering methods, including regression-based, classification-based, and decomposition-based clustering tasks.

---

### Meta-Review · Area_Chair_VmDs · 2026-01-13

**Summary:**

Reviewer JabB concerns its limited analysis on computational overhead and scalability as well as dual-label interaction. Reviewer pnEa concerns its model complexity and limited benchmark datasets. Reviewer kJk3  mainly concerns the model novelty and computational cost.
Reviewer QtwU concerns the experimental credibility, fairness, insufficiency.

**Reviewer Concerns:**

After rebuttal, the potential limitations, hyperparameters strategies, computational overhead, and running time have been discussed. However, the proposed model is over-complex, and insufficient innovation and limited benchmark datasets further weaken the contributions.

**Reviewer Scores:**

After rebuttal, Reviewer JabB may raise the score from 6 to 8, as the main limitations, hyperparameters-tuning strategies, dual-label function have been discussed. Reviewers pnEa, kJk3 and QtwU may maintain the original score due to model complexity, limited benchmark datasets, computational burden, unconvincing results.

---

### Decision · Program_Chairs · 2026-01-26

Reject